# $\alpha$-DIVERGENCE LOSS FUNCTION FOR NEURAL DENSITY RATIO ESTIMATION

## ABSTRACT

Density ratio estimation (DRE) is a fundamental machine learning technique for capturing relationships between two probability distributions. State-of-the-art DRE methods estimate the density ratio using neural networks trained with loss functions derived from variational representations of $f$-divergence. However, existing methods face optimization challenges, such as overfitting due to lower-unbounded loss functions, biased mini-batch gradients, vanishing training loss gradients, and high sample requirements for Kullback-Leibler (KL) divergence loss functions. To address these issues, we focus on $\alpha$-divergence, which provides a suitable variational representation of $f$-divergence. Subsequently, a novel loss function for DRE, the $\alpha$-divergence loss function ($\alpha$-Div), is derived. $\alpha$-Div is concise but offers stable and effective optimization for DRE. The boundedness of $\alpha$-divergence provides the potential for successful DRE with data exhibiting high KL-divergence. Our numerical experiments demonstrate the effectiveness in optimization using $\alpha$-Div. However, the experiments also show that the proposed loss function offers no significant advantage over the KL-divergence loss function in terms of RMSE for DRE. This indicates that the accuracy of DRE is primarily determined by the amount of KL-divergence in the data and is less dependent on $\alpha$-divergence.

## 1 INTRODUCTION

Density ratio estimation (DRE), a fundamental technique in various machine learning domains, estimates the density ratio $r^*(\mathbf{x}) = q(\mathbf{x})/p(\mathbf{x})$ between two probability densities using two sample sets drawn separately from $p$ and $q$. Several machine learning methods, including generative modeling (Goodfellow et al., 2014; Nowozin et al., 2016; Uehara et al., 2016), mutual information estimation and representation learning (Belghazi et al., 2018; Hjelm et al., 2018), energy-based modeling (Gutmann & Hyvärinen, 2010), and covariate shift and domain adaptation (Shimodaira, 2000; Huang et al., 2006), involve problems where DRE is applicable. Given its potential to enhance a wide range of machine learning methods, the development of effective DRE techniques has garnered significant attention.

Recently, neural network-based methods for DRE have achieved state-of-the-art results. These methods train neural networks as density ratio functions using loss functions derived from variational representations of $f$-divergence (Nguyen et al., 2010), which are equivalent to density-ratio matching under Bregman divergence (Sugiyama et al., 2012). The optimal function for a variational representation of $f$-divergence, through the Legendre transform, corresponds to the density ratio.

However, existing neural network methods suffer from several issues. First, an overfitting phenomenon, termed *train-loss hacking* by Kato & Teshima (2021), occurs during optimization when lower-unbounded loss functions are used. Second, the gradients of loss functions over mini-batch samples provide biased estimates of the full gradient when using standard loss functions derived directly from the variational representation of $f$-divergences (Belghazi et al., 2018). Third, loss function gradients can vanish when the estimated probability ratios approach zero or infinity (Arjovsky & Bottou, 2017). Finally, optimization with a Kullback–Leibler (KL)-divergence loss function often fails on high KL-divergence data because the sample requirement for optimization increases exponentially with the true amount of KL-divergence (Poole et al., 2019; Song & Ermon, 2019; McAllester & Stratos, 2020).

To address these problems, this study focuses on $\alpha$-divergence, a subgroup of $f$-divergence, which has a sample complexity independent of its ground truth value. We then present a Gibbs density representation for a variational form of the divergence to obtain unbiased mini-batch gradients, from which we derive a novel loss function for DRE, referred to as the $\alpha$-divergence loss function ($\alpha$-Div). Despite its conciseness, $\alpha$-Div offers stable and effective optimization for DRE.

Furthermore, this study provides technical justifications for the proposed loss function. $\alpha$-Div has a sample complexity that is independent of the ground truth value of $\alpha$-divergence and provides unbiased mini-batch gradients of training losses. Additionally, $\alpha$-Div is lower-bounded when $\alpha$ is within a specific interval, thereby preventing train-loss hacking during optimization. By selecting $\alpha$ from this interval, we also avoid vanishing gradients in neural networks when they reach extreme local minima. We empirically validate our approach through numerical experiments using toy datasets, which demonstrate the stability and efficiency of the proposed loss function during optimization.

However, we observe that the root mean squared error (RMSE) of the estimated density ratios increases significantly for data with higher KL-divergence when using the proposed loss function. The same phenomenon is observed with the KL-divergence loss function. These results suggest that the accuracy of DRE is primarily determined by the amount of KL-divergence in the data and is less influenced by the $\alpha$-divergence, providing insights into the accuracy of downstream tasks in DRE using $f$-divergence loss functions.

The key contributions of this study are as follows: (1) A novel loss function for DRE, $\alpha$-Div, is proposed, offering a concise solution to the instability and biased gradient issues in existing $f$-divergence loss functions. (2) Technical justifications for the proposed loss function are presented. (3) We empirically confirm the stability and efficiency of the proposed loss function in optimization. (4) We find that RMSE in DRE increases significantly as the KL-divergence in the data rises when using the $\alpha$-divergence loss function, indicating that DRE and KL-divergence estimation accuracy is primarily determined by the amount of KL-divergence in the data, rather than the $\alpha$-divergence.

## 2  PROBLEM SETUP

**Problem definition.**  $P$ and $Q$ are probability distributions on $\Omega \subset \mathbb{R}^d$ with unknown probability densities $p$ and $q$, respectively. We assume $p(\mathbf{x}) > 0 \Leftrightarrow q(\mathbf{x}) > 0$ at almost everywhere $\mathbf{x} \in \Omega$. The goal of DRE is to accurately estimate $r^*(\mathbf{x}) = q(\mathbf{x})/p(\mathbf{x})$ from given i.i.d. samples $\hat{\mathbf{X}}_{P[R]} = \{\mathbf{x}_i^p\}_{i=1}^R \sim p$ and $\hat{\mathbf{X}}_{Q[S]} = \{\mathbf{x}_i^q\}_{i=1}^S \sim q$.

**Additional notation.**  $E_P[\cdot]$ denotes an expectation under the distribution $P$: $E_P[\phi(\mathbf{x})] = \int_\Omega \phi(\mathbf{x})dP(\mathbf{x})$, where $\phi(\mathbf{x})$ is a measurable function over $\Omega$. $\hat{E}_{P[R]}[\cdot]$ denotes the empirical expectation of $\hat{\mathbf{X}}_{P[R]}$: $\hat{E}_{P[R]}[\phi(\mathbf{x})] = \sum_{i=1}^R \phi(\mathbf{x}_i^p)/R$. Variables of a function or the superscript variable " $[R]$ " of $\hat{E}_{P[R]}$ are omitted when unnecessary and represented as $E_P[\phi]$ or $\hat{E}_P[\phi]$. Similarly, notations $E_Q[\cdot]$ and $\hat{E}_{Q[S]}[\cdot]$ are defined. $E[\cdot]$ is written for $E_P[E_Q[\cdot]]$.

## 3  DRE VIA $f$-DIVERGENCE VARIATIONAL REPRESENTATIONS AND ITS MAJOR PROBLEMS

In this section, we introduce DRE using $f$-divergence variational representations and $f$-divergence loss functions. First, we review the definition of $f$-divergences. Next, we identify four major issues with existing $f$-divergence loss functions: the overfitting problem with lower-unbounded loss functions, biased mini-batch gradients, vanishing training loss gradients, and high sample requirements for Kullback-Leibler (KL) divergence loss functions.

### 3.1  DRE VIA $f$-DIVERGENCE VARIATIONAL REPRESENTATION

First, we review the definition of $f$-divergences.

**Definition 3.1** ($f$-divergence).  The $f$-divergence $D_f$ between two probability measures $P$ and $Q$, which is induced by a convex function $f$ satisfying $f(1) = 0$, is defined as $D_f(Q||P) = E_P[f(q(\mathbf{x})/p(\mathbf{x}))]$.

Many divergences are specific cases obtained by selecting a suitable generator function $f$. For example, $f(u) = u \cdot \log u$ corresponds to KL-divergence.

Then, we derive the variational representations of $f$-divergences using the Legendre transform of the convex conjugate of a twice differentiable convex function $f$, $f^*(\psi) = \sup_{u \in \mathbb{R}} \{\psi \cdot u - f(u)\}$ (Nguyen et al., 2007):

$$D_f(Q||P) = \sup_{\phi \geq 0} \left\{ E_Q[f'(\phi)] - E_P[f^*(f'(\phi))] \right\}, \tag{1}$$

where the supremum is taken over all measurable functions $\phi : \Omega \to \mathbb{R}$ with $E_Q[|f'(\phi)|] < \infty$ and $E_P[|f^*(f'(\phi))|] < \infty$. The maximum value is achieved at $\phi(\mathbf{x}) = q(\mathbf{x})/p(\mathbf{x})$.

By replacing $\phi$ with a neural network model $\phi_\theta$, the optimal function for Equation (1) is trained through back-propagation using an $f$-divergence loss function, such that

$$\mathcal{L}_f^{(R,S)}(\phi_\theta) = - \left\{ \hat{E}_{Q[S]}[f'(\phi_\theta)] - \hat{E}_{P[R]}[f^*(f'(\phi_\theta))] \right\}, \tag{2}$$

where $\phi_\theta$ is a real-valued function, the superscript variable " $(R, S)$ " is omitted when unnecessary and represented as $\mathcal{L}_f(\cdot)$. As shown in Table 1, we list pairs of convex functions and the corresponding loss functions $\mathcal{L}_f(\phi_\theta)$ in Equation (2) for several $f$-divergences.

## 3.2 TRAIN-LOSS HACKING PROBLEM

When $f$-divergence loss functions $\mathcal{L}_f(\phi_\theta)$, as defined in Equation (2), are not lower-bounded, over-fitting can occur during optimization. For example, in the case of the Pearson $\chi^2$ loss function, $\mathcal{L}_{\text{chi-sq}} = -2 \cdot \hat{E}_Q[\phi_\theta] + \hat{E}_P[\phi_\theta^2]$, overfitting occurs as follows. Since the term $-2 \cdot \hat{E}_Q[\phi_\theta]$ is not lower-bounded, it can approach negative infinity, causing the entire loss function to diverge to negative infinity as $\phi_\theta(\mathbf{x}_i^q) \to \infty$ for $\mathbf{x}_i^q \in \hat{\mathbf{X}}_{Q[S]}$. Therefore, $\mathcal{L}_{\text{chi-sq}} \to -\infty$ when $\phi_\theta(\mathbf{x}_i^q) \to \infty$ for some $\mathbf{x}_i^q \in \hat{\mathbf{X}}_{Q[S]}$. As shown in Table 1, both the KL-divergence and Pearson $\chi^2$ loss functions are not lower-bounded, and hence, are prone to overfitting during optimization. This phenomenon is referred to as *train-loss hacking* by Kato & Teshima (2021).

## 3.3 BIASED GRADIENT PROBLEM

Neural network parameters are optimized by summarizing the gradients of the loss function for each mini-batch. It is desirable for these gradients to be unbiased, i.e., $E[\nabla_\theta \mathcal{L}_f(\theta)] = \nabla_\theta E[\mathcal{L}_f(\theta)]$ holds. However, standard $f$-divergence loss functions derived solely from Equation (2) often result in biased gradients. To illustrate this, consider the case of a KL-divergence loss function, where $f(u) = u \cdot \log u$. The loss function is given by $\mathcal{L}_{KL}(\phi_\theta) = -\hat{E}_Q[\log \phi_\theta] + \hat{E}_P[\phi_\theta] - 1$, and the gradient is expressed as $\nabla_\theta \mathcal{L}_{KL}(\phi_\theta) = -\hat{E}_Q[\nabla_\theta(\log \phi_\theta)] + \hat{E}_P[\nabla_\theta(\phi_\theta)]$. Notably, it does not hold that $\nabla_\theta E_Q[\log \phi_\theta] = E_Q[\nabla_\theta(\log \phi_\theta)]$ because $E_Q[\cdot]$ represents an integral over $\Omega$. For example, consider $\phi_\theta = |x - \theta|$, where $x \in [0, 1]$ and $\theta \in (0, 1)$. Then, $\frac{\partial}{\partial \theta} E[\log \phi_\theta(x)] = \frac{\partial}{\partial \theta} \int_0^1 \log |x - \theta| \, dx = -\log(1 - \theta)$, whereas $E[\frac{\partial}{\partial \theta} \log \phi_\theta(x)] = \int_0^\theta \frac{1}{\theta - x} dx + \int_\theta^1 \frac{1}{x - \theta} dx = \infty$.

Conversely, this equality does hold for the sample mean $\hat{E}_Q$, as shown above. Therefore, we find that $\nabla_\theta E[\mathcal{L}_{KL}(\phi_\theta)] \neq E[\nabla_\theta \mathcal{L}_{KL}(\phi_\theta)]$. To mitigate this issue, Belghazi et al. (2018) introduced a bias-reduction method for stochastic gradients in KL-divergence loss functions.

## 3.4 VANISHING GRADIENTS PROBLEM

The vanishing-gradient problem in optimizing divergences is a well-known issue in GANs (Arjovsky & Bottou, 2017). To address the vanishing gradients of training losses, loss functions with penalty terms have been proposed (Gulrajani et al., 2017; Roth et al., 2017).

We believe this problem occurs when the following two conditions are satisfied: (i) The loss function causes slight updates to the model parameters, and (ii) Updating the model parameters causes minimal changes in the model outputs. Therefore, the problem can arise when the following relations hold:

$$\underbrace{E\big[\nabla_\theta \mathcal{L}_f(\phi_\theta)\big] = \vec{0}}_{\text{(i)}} \quad \& \quad \underbrace{E_Q\big[\nabla_\theta \phi_\theta\big] = \vec{0} \;\&\; E_P\big[\nabla_\theta \phi_\theta\big] = \vec{0}}_{\text{(ii)}}, \tag{3}$$

where $\vec{0}$ denotes a vector of zeros the same length as the model gradient.

In Equation (3), (i) represents the vanishing gradient of the loss function. Notably, (i) does not necessarily lead to (ii). For example, in the case of the KL divergence loss, the gradient is expressed as $E\big[\nabla_\theta \mathcal{L}_{KL}(\phi_\theta)\big] = -E_Q\big[\nabla_\theta \phi_\theta / \phi_\theta\big] + E_P\big[\nabla_\theta \phi_\theta\big]$. Then, we observe that (ii) generally cannot be derived from $E\big[\nabla_\theta \mathcal{L}_{KL}(\phi_\theta)\big] = \vec{0}$. Additionally, (ii) serves as a condition that ensures (i) persists. In fact, since (ii) is equivalent to the condition where no updates of the model parameters occur, the model parameters are not updated under (ii). Consequently, the model's predictions do not change, yielding the same results as the current step. Thus, when (i) and (ii) are satisfied, the loss gradient remains near zero.

Now, consider a case where the estimated density ratio becomes either very small or very large, leading to sufficient conditions for Equation (3) to hold. Table 2 lists the gradient formulas for divergence loss functions from Table 1, along with the asymptotic behavior of the loss gradients as $\phi_\theta \to 0$ or $\phi_\theta \to \infty$. These results demonstrate that major $f$-divergence loss functions satisfy the conditions for Equation (3), such that $E\big[\nabla_\theta \mathcal{L}_f(\phi_\theta)\big] \to c_1 \cdot E_Q\big[\nabla_\theta \phi_\theta\big] + c_2 \cdot E_P\big[\nabla_\theta \phi_\theta\big]$, where $c_1$ and $c_2$ are constants, as $\phi_\theta \to 0$ or $\phi_\theta \to \infty$.

In summary, all the divergence loss functions in Tables 1 and 2 can experience vanishing gradients when the estimated density ratio approaches extreme local minima.

### 3.5 SAMPLE SIZE REQUIREMENT PROBLEM FOR KL-DIVERGENCE

The sample complexity of the KL-divergence is $O(e^{KL(Q||P)})$, which implies that

$$\lim_{N \to \infty} N \cdot \mathrm{Var}\Big[\widehat{KL^N}(Q||P)\Big] \geq e^{KL(Q||P)} - 1, \tag{4}$$

where $\widehat{KL^N}(Q||P)$ represents an arbitrary KL-divergence estimator for sample size $N$ using a variational representation of the divergence, and $KL(Q||P)$ represents the true value of KL-divergence (Poole et al., 2019; Song & Ermon, 2019; McAllester & Stratos, 2020). That is, when using KL-divergence loss functions, the sample size of the training data must increase exponentially as the true amount of KL-divergence increases in order to sufficiently train a neural network. To address this issue, existing methods divide the estimation of high divergence values into multiple smaller divergence estimations (Rhodes et al., 2020).

## 4 DRE USING A NEURAL NETWORK WITH AN $\alpha$-DIVERGENCE LOSS

In this section, we derive our loss function from a variational representation of $\alpha$-divergence and present the training and prediction methods using this loss function. The exact claims and proofs for all theorems are provided in C.2 in the Appendix.

### 4.1 DERIVATION OF OUR LOSS FUNCTION FOR DRE

Here, we define $\alpha$-divergence (Amari's $\alpha$-divergence), which is a subgroup of $f$-divergence, as (Amari & Nagaoka, 2000):

$$D_\alpha(Q||P) = E_P\left[\frac{1}{\alpha \cdot (\alpha - 1)} \cdot \left\{\left(\frac{q(\mathbf{x})}{p(\mathbf{x})}\right)^{1-\alpha} - 1\right\}\right], \tag{5}$$

where $\alpha \in \mathbb{R} \setminus \{0, 1\}$. From Equation (5), Hellinger divergence is obtained when $\alpha = 1/2$, and $\chi^2$ when $\alpha = -1$.

Then, we achieve the following variational representation of $\alpha$-divergence :

**Theorem 4.1.** *A variational representation of $\alpha$-divergence is given as*

$$D_\alpha(Q||P) = \sup_{\phi \geq 0} \left\{\frac{1}{\alpha \cdot (1-\alpha)} - \frac{1}{\alpha} \cdot E_Q\Big[\phi^\alpha\Big] - \frac{1}{1-\alpha} \cdot E_P\Big[\phi^{\alpha-1}\Big]\right\}, \tag{6}$$

Table 1: List of $f$-divergence loss functions $\mathcal{L}_f(\phi_\theta)$ in Equation (2) their associated convex functions, and their lower-boundedness status. Part of the list of divergences and their convex functions is based on Nowozin et al. (2016).

| Name | convex function $f$ | $\mathcal{L}_f(\phi_\theta)$ | Lower-bounded? |
|------|---------------------|------------------------------|----------------|
| KL | $u \cdot \log u$ | $-\hat{E}_Q\big[\log(\phi_\theta)\big] + \hat{E}_P\big[\phi_\theta\big] - 1$ | No |
| Pearson $\chi^2$ | $(u-1)^2$ | $-2 \cdot \hat{E}_Q\big[\phi_\theta\big] + \hat{E}_P\big[\phi_\theta^2\big]$ | No |
| Squared Hellinger | $(\sqrt{u}-1)^2$ | $\hat{E}_Q\big[\phi_\theta^{-1/2}\big] + \hat{E}_P\big[\phi_\theta^{1/2}\big] - 2$ | Yes |
| GAN | $u \cdot \log u$ $-(u+1)\log(u+1)$ | $\hat{E}_Q\big[\log(1+\phi_\theta^{-1})\big]$ $+ \hat{E}_P\big[\log(1+\phi_\theta)\big]$ | Yes |

Table 2: List of gradient formulas $\nabla_\theta \mathcal{L}_f(\phi_\theta)$ of loss functions $\mathcal{L}_f(\phi_\theta)$ in Table 1 and the asymptotic behavior of $E\big[\nabla_\theta \mathcal{L}_f(\phi_\theta)\big]$ as $\phi_\theta \to 0$ or $\phi_\theta \to \infty$ under regular conditions. A symbol " $*$ " in the table indicates that the asymptotic value cannot be expressed as a linear combination of $E_Q\big[\nabla_\theta \phi_\theta\big]$ and $E_P\big[\nabla_\theta \phi_\theta\big]$, and $\vec{0}$ denotes a vector of zeros with the same length as the model gradient.

| Name | $\nabla_\theta \mathcal{L}_f(\phi_\theta)$ | $E\big[\nabla_\theta \mathcal{L}_f(\phi_\theta)\big] \to ?$ | |
|------|---------------------------------------------|-----------------------------------------------------------|---|
| | | $\phi_\theta \to 0$ | $\phi_\theta \to \infty$ |
| KL | $-\hat{E}_Q\big[\nabla_\theta \phi_\theta / \phi_\theta\big] + \hat{E}_P\big[\nabla_\theta \phi_\theta\big]$ | $*$ | $E_Q\big[\nabla_\theta \phi_\theta\big]$ |
| Pearson $\chi^s$ | $-2 \cdot \hat{E}_Q\big[\nabla_\theta \phi_\theta\big] + 2 \cdot \hat{E}_P\big[\nabla_\theta \phi_\theta \cdot \phi_\theta\big]$ | $-2 \cdot E_Q\big[\nabla_\theta \phi_\theta\big]$ | $*$ |
| Squared Hellinger | $-\frac{1}{2} \cdot \hat{E}_Q\big[\nabla_\theta \phi_\theta \cdot \phi_\theta^{-3/2}\big]$ $+ \frac{1}{2} \cdot \hat{E}_P\big[\nabla_\theta \phi_\theta \cdot \phi_\theta^{-1/2}\big]$ | $*$ | $\vec{0}$ |
| GAN | $-\hat{E}_Q\big[\nabla_\theta \phi_\theta / (1+\phi_\theta)\big]$ $+ \hat{E}_P\big[\nabla_\theta \phi_\theta / (1+\phi_\theta)\big]$ | $-E_Q\big[\nabla_\theta \phi_\theta\big]$ $+ E_P\big[\nabla_\theta \phi_\theta\big]$ | $\vec{0}$ |

*where the supremum is taken over all overall measurable functions satisfying $E_P[\phi^{1-\alpha}] < \infty$ and $E_Q[\phi^{-\alpha}] < \infty$. The maximum value is achieved at $\phi(\mathbf{x}) = q(\mathbf{x})/p(\mathbf{x})$.*

From the right-hand side of Equation (6), we obtain a standard $\alpha$-divergence loss function as

$$\mathcal{L}_{\alpha\text{-standard}}^{(R,S)}(\phi_\theta \,;\, \alpha) = \frac{1}{\alpha} \cdot \hat{E}_Q\Big[\phi_\theta^\alpha\Big] + \frac{1}{1-\alpha} \cdot \hat{E}_P\Big[\phi_\theta^{\alpha-1}\Big]. \tag{7}$$

The above loss function has a biased gradient because $\nabla_\theta E_Q\big[\phi_\theta^\alpha\big] \neq E_Q\big[\nabla_\theta(\phi_\theta^\alpha)\big]$ and $\nabla_\theta E_P\big[\phi_\theta^{\alpha-1}\big] \neq E_P\big[\nabla_\theta(\phi_\theta^{\alpha-1})\big]$ are generally observed. To obtain unbiased gradients for our function, we rewrite the terms $\phi_\theta^\alpha$ and $\phi_\theta^{\alpha-1}$ of the equation in Gibbs density form. Then, we have another variational representation of $\alpha$-divergence as

**Theorem 4.2.** *A variational representation of $\alpha$-divergence is given as*

$$D_\alpha(Q||P) = \sup_{T:\Omega \to \mathbb{R}} \left\{ \frac{1}{\alpha \cdot (1-\alpha)} - \frac{1}{\alpha} \cdot E_Q\Big[e^{\alpha \cdot T}\Big] - \frac{1}{1-\alpha} \cdot E_P\Big[e^{(\alpha-1) \cdot T}\Big] \right\}, \tag{8}$$

*where the supremum is taken over all measurable function $T : \Omega \to \mathbb{R}$ satisfying $E_P[e^{(\alpha-1) \cdot T}] < \infty$ and $E_Q[e^{\alpha \cdot T}] < \infty$. The equality holds for $T^*$ satisfying $e^{-T^*(\mathbf{x})} = q(\mathbf{x})/p(\mathbf{x})$.*

Subsequently, we obtain our loss function for DRE, called $\alpha$-Divergence loss function ($\alpha$-Div).

**Definition 4.3** ($\alpha$-Div)**.** $\alpha$-Divergence loss is defined as:

$$\mathcal{L}_{\alpha\text{-Div}}^{(R,S)}(T_\theta \,;\, \alpha) = \frac{1}{\alpha} \cdot \hat{E}_{Q[S]}\Big[e^{\alpha \cdot T_\theta}\Big] + \frac{1}{1-\alpha} \cdot \hat{E}_{P[R]}\Big[e^{(\alpha-1) \cdot T_\theta}\Big]. \tag{9}$$

The superscript " $(R, S)$ " is dropped when unnecessary and is given as $\mathcal{L}_{\alpha\text{-Div}}(T_\theta \,;\, \alpha)$.

---

**Algorithm 1** Training for DRE with $\alpha$-Div

---

**Input:** Data from a denominator distribution $\{\mathbf{x}_i^p\}_{i=1}^R$, and from a numerator distribution $\{\mathbf{x}_i^q\}_{i=1}^S$.

**Output:** A Neural Network Model $T_{\theta_N}$.

**for** $t = 1$ to $N$ **do**
$$\hat{E}_P \leftarrow \frac{1}{R} \sum_{i=1}^R e^{(\alpha-1) \cdot T_{\theta_t}(\mathbf{x}_i^p)}$$
$$\hat{E}_Q \leftarrow \frac{1}{S} \sum_{i=1}^S e^{\alpha \cdot T_{\theta_t}(\mathbf{x}_i^q)}$$
$$\mathcal{L}_{\alpha\text{-Div}}(\theta_t) \leftarrow \frac{\hat{E}_Q}{\alpha} + \frac{\hat{E}_P}{1-\alpha}$$
$$\theta_{t+1} \leftarrow \theta_t - \nabla_{\theta_t} \mathcal{L}_{\alpha\text{-Div}}(\theta_t)$$
**end for**

---

## 4.2 Training and predicting with $\alpha$-Div

We train a neural network with $\alpha$-Div as described in Algorithm 1. In practice, neural networks rarely achieve the maximum in Equation (8). The following theorem suggests that normalizing the estimated values, $q(\mathbf{x})/p(\mathbf{x}) = e^{-T_\theta(\mathbf{x})}/\hat{E}_P\left[e^{-T_\theta}\right]$, improves the optimization of the neural networks.

**Theorem 4.4.** *For a fixed function $T : \Omega \to \mathbb{R}$, let $c^*$ be the optimal scalar value for the following infimum:*

$$c^* = \arg\inf_{c \in \mathbb{R}} E\left[\mathcal{L}_{\alpha\text{-}Div}(T-c)\right] = \arg\inf_{c \in \mathbb{R}} \left\{ \frac{1}{\alpha} \cdot E_Q\left[e^{\alpha \cdot (T-c)}\right] + \frac{1}{1-\alpha} \cdot E_P\left[e^{(\alpha-1) \cdot (T-c)}\right] \right\}. \tag{10}$$

*Then, $c^*$ satisfies $E_P\left[e^{-T-c^*}\right] = 1$. That is, $e^{-T-c^*} = e^{-T}/E_P\left[e^{-T}\right]$.*

# 5 Theoretical results for the proposed loss function

In this section, we provide theoretical results that justify our approach with $\alpha$-Div. The exact claims and proofs for all the theorems are provided in Section C.3 in the Appendix.

## 5.1 Addressing the train-loss hacking problem

$\alpha$-Div avoids the train-loss hacking problem when $\alpha$ is within $(0, 1)$. Table 3 summarizes the lower-boundedness status of $\alpha$-Div for each case: $\alpha < 0$, $0 < \alpha < 1$, or $\alpha > 1$. $\alpha$-Div is lower-bounded when $0 < \alpha < 1$, whereas it is not lower-bounded when $\alpha > 1$ or $\alpha < 0$. Thus, the train-loss hacking problem is avoided when $\alpha$ is selected from the interval $(0, 1)$.

## 5.2 Unbiasedness of gradients

The following Theorem 5.1 guarantees the unbiasedness of the gradients of $\alpha$-Div.

**Theorem 5.1** (Brief and informal). *Let $T_\theta(\mathbf{x}) : \Omega \to \mathbb{R}$ be a function such that the map $\theta = (\theta_1, \theta_2, \ldots, \theta_p) \in \Theta \mapsto T_\theta(\mathbf{x})$ is differentiable for all $\theta$ and $\mu$-almost every $\mathbf{x} \in \Omega$. Assume some regular conditions, including the local Lipschitz continuity of $T_\theta$. Then, the following holds:*

$$E\left[\nabla_\theta \mathcal{L}_{\alpha\text{-}Div}(T_\theta; \alpha)\big|_{\theta=\bar{\theta}}\right] = \nabla_\theta E\left[\mathcal{L}_{\alpha\text{-}Div}(T_\theta; \alpha)\right]\big|_{\theta=\bar{\theta}}. \tag{11}$$

## 5.3 Addressing gradient vanishing problem

When $\alpha$ is within $(0, 1)$, $\alpha$-Div avoids the vanishing gradient problem in its training losses. Below, we describe why gradient vanishing does not occur in this case.

First, we obtain the gradients of the standard $\alpha$-divergence loss in Equation (7) and $\alpha$-Div:

$$\nabla_\theta \mathcal{L}_{\alpha\text{-standard}}(\phi_\theta) = \hat{E}_Q\left[\nabla_\theta \phi_\theta \cdot \phi_\theta^{\alpha-1}\right] - \hat{E}_P\left[\nabla_\theta \phi_\theta \cdot \phi_\theta^{-\alpha}\right], \tag{12}$$

$$\nabla_\theta \mathcal{L}_{\alpha\text{-Div}}(e^{T_\theta}) = \hat{E}_Q\left[\nabla_\theta T_\theta \cdot e^{\alpha \cdot T_\theta}\right] - \hat{E}_P\left[\nabla_\theta T_\theta \cdot e^{(\alpha-1) \cdot T_\theta}\right]. \tag{13}$$

Table 3: Lower-boundedness status of $\alpha$-Div for each case of $\alpha < 0$, $\alpha > 1$, and $0 < \alpha < 1$.

| Intervals of $\alpha$ | $\frac{1}{\alpha} \cdot \hat{E}_{Q[S]}$ | $+$ | $\frac{1}{1-\alpha} \cdot \hat{E}_P\left[e^{(\alpha-1)\cdot T}\right]$ | $=$ | $\mathcal{L}_{\alpha\text{-Div}}(T;\alpha)$ |
|---|---|---|---|---|---|
| $\alpha < 0$ | $\downarrow -\infty$ (as $e^T \uparrow \infty$) | | $\geq 0$ | | lower-unbounded |
| $\alpha > 1$ | $\geq 0$ | | $\downarrow -\infty$ (as $e^T \uparrow \infty$) | | lower-unbounded |
| $0 < \alpha < 1$ | $\geq 0$ | | $\geq 0$ | | **lower-bounded** |

Table 4: Behavior of $E\left[\nabla_\theta \mathcal{L}_{\alpha\text{-standard}}(\phi_\theta)\right]$ and $E\left[\nabla_\theta \mathcal{L}_{\alpha\text{-Div}}(T_\theta)\right]$ as estimated probability ratios approach 0 or $\infty$, for each case of $\alpha < 0$, $\alpha > 1$, and $0 < \alpha < 1$.

| | $\left\|E\left[\nabla_\theta \mathcal{L}_{\alpha\text{-standard}}(\phi_\theta)\right]\right\|_\infty \to ?$ | | $\left\|E\left[\nabla_\theta \mathcal{L}_{\alpha\text{-Div}}(T_\theta)\right]\right\|_\infty \to ?$ | |
|---|---|---|---|---|
| Intervals of $\alpha$ | $E_P[\phi_\theta] \to 0$ | $E_P[\phi_\theta] \to \infty$ | $E_P[e^{T_\theta}] \to 0$ | $E_P[e^{T_\theta}] \to \infty$ |
| $\alpha < 0$ | $\infty$ | **0** | $\infty - \infty$ | **0** |
| $\alpha > 1$ | **0** | $-\infty$ | **0** | $\infty - \infty$ |
| $0 < \alpha < 1$ | $\infty$ | **0** | $-\infty$ | $\infty$ |

Next, consider the case where the estimated probability ratios, $\phi_\theta$ and $e^{T_\theta}$, are either nearly zero or very large for some point $\mathbf{x}$. Let $\left\|\theta\right\|_\infty$ denote the maximum value of all elements in $\theta$. Under the assumption that $p(\mathbf{x}) > 0 \Leftrightarrow q(\mathbf{x}) > 0$ for all $\mathbf{x} \in \Omega$, we observe the following equivalences: $E_Q[e^{T_\theta}] \to 0 \Leftrightarrow E_P[e^{T_\theta}] \to 0$; and $E_Q[e^{T_\theta}] \to \infty \Leftrightarrow E_P[e^{T_\theta}] \to \infty$.

Finally, the behavior of $E\left[\nabla_\theta \mathcal{L}_{\alpha\text{-Div}}(T_\theta;\alpha)\right]$ when $E_P[e^{T_\theta}] \to 0$ or $E_P[e^{T_\theta}] \to \infty$, under certain regular conditions for $T_\theta$, is summarized in Table 4.

In all cases except for $\alpha$-Div with $0 < \alpha < 1$, vanishing of the loss gradients is observed, such that $\left\|E\left[\nabla_\theta \mathcal{L}_{\alpha\text{-standard}}(\phi_\theta)\right]\right\|_\infty \to 0$ or $\left\|E\left[\nabla_\theta \mathcal{L}_{\alpha\text{-Div}}(T_\theta)\right]\right\|_\infty \to 0$. This implies that, during optimization, neural networks may remain stuck at extreme local minima when their estimations for density ratios are either 0 or $\infty$. However, this issue is avoided when $\alpha$ is within the interval $(0,1)$. Additionally, choosing $\alpha$ within the interval $(0,1)$ avoids numerical instability in cases where $\left\|E\left[\nabla_\theta \mathcal{L}_{\alpha\text{-Div}}(T_\theta;\alpha)\right]\right\|_\infty \to \infty - \infty$, which occurs for $\alpha > 1$ and $\alpha < 0$.

## 5.4 Sample size requirements for optimizing $\alpha$-divergence

We present the exact upper bound on the sample size required for minimizing $\alpha$-Div in Theorem 5.2, which corresponds to Equation (4) for KL-divergence loss functions. The sample size requirement for minimizing $\alpha$-Div is upper-bounded depending on the value of $\alpha$. Intuitively, this property arises from the boundedness of Amari's $\alpha$-divergence: $0 \leq D_\alpha \leq 1/(\alpha \cdot (1-\alpha))$.

**Theorem 5.2.** *Let* $T^* = -\log(q(\mathbf{x})/p(\mathbf{x}))$ *and* $N = \min\{R, S\}$. *Subsequently, let*

$$\hat{D}^{(N)}(Q\|P\,;\alpha) = \frac{1}{\alpha \cdot (1-\alpha)} - \mathcal{L}_{\alpha\text{-Div}}^{(N,N)}(T^*\,;\alpha). \tag{14}$$

*Then,*

$$\sqrt{N} \cdot \left\{\hat{D}^{(N)}(Q\|P\,;\alpha) - D(Q\|P\,;\alpha)\right\} \xrightarrow{d} \mathcal{N}(0,\sigma_\alpha) \tag{15}$$

*holds, where*

$$\sigma_\alpha^2 = C_\alpha^1 \cdot D(Q\|P\,;2\alpha) + C_\alpha^2 \cdot D(Q\|P\,;2\alpha-1)$$
$$+ C_\alpha^3 \cdot D(Q\|P\,;\alpha)^2 + C_\alpha^4 \cdot D(Q\|P\,;\alpha) + C_\alpha^5, \tag{16}$$

*and* $C_\alpha^1 = 2\alpha \cdot (1-2\alpha)/\alpha^2$, $C_\alpha^2 = 2\alpha \cdot (1-2\alpha)/(1-\alpha)^2$, $C_\alpha^3 = -1/\alpha^2 - 1/(1-\alpha)^2$, $C_\alpha^4 = 2/\alpha^2 + 2/(1-\alpha)^2$, *and* $C_\alpha^5 = (1/\alpha^2 + 1/(1-\alpha)^2) \cdot (2 - 2\alpha \cdot (1-\alpha))$.

Unfortunately, despite the sample requirement stated in Equation (15), we empirically find that the estimation accuracy for $\alpha$-Div and KL-divergence loss functions is roughly the same in downstream tasks of DRE, including KL-divergence estimation, as discussed in Section 6.3.

## 6 EXPERIMENTS

We evaluated the performance of our approach using synthetic datasets. First, we assessed the stability of the proposed loss function due to its lower-boundedness for $\alpha$ within $(0, 1)$. Second, we validated the effectiveness of our approach in addressing biased gradients in the training losses. Finally, we examined the $\alpha$-divergence loss function for DRE using high KL-divergence data. Details on the experimental settings and neural network training are provided in Section D in the Appendix.

In addition to the results presented in this section, we conducted two additional experiments: a comparison of $\alpha$-Div with existing DRE methods, and experiments using real-world data. These additional experiments are reported in Section E in the Appendix.

### 6.1 EXPERIMENTS ON THE STABILITY OF OPTIMIZATION FOR DIFFERENT VALUES OF $\alpha$

We empirically confirmed the stability of optimization using $\alpha$-Div, as discussed in Section 5.1. This includes addressing the potential divergence of training losses for $\alpha > 1$ and $\alpha < 0$, and observing the stability of optimization when $\alpha$ is within $(0, 1)$. Subsequently, we conducted experiments using synthetic datasets to examine the behavior of training losses during optimization across different values of $\alpha$ at each learning step.

**Experimental Setup.** First, we generated 100 training datasets from two 5-dimensional normal distributions, $P = \mathcal{N}(\mu_p, I_5)$ and $Q = \mathcal{N}(\mu_q, \Sigma_q)$, where $\mu_p = \mu_q = (0, 0, \ldots, 0)$, and $I_5$ denotes the 5-dimensional identity matrix. The covariance matrix $\Sigma_q = (\sigma_{ij})_{i=1}^5$ is defined as $\sigma_{ii} = 1$, and $\sigma_{ij} = 0.8$ for $i \neq j$. Subsequently, we trained neural networks using the synthetic datasets by optimizing $\alpha$-Div for $\alpha = -3.0, -2.0, -1.0, 0.2, 0.5, 0.8, 2.0, 3.0$, and $4.0$, while measuring training losses at each learning step. For each value of $\alpha$, 100 trials were performed. Finally, we reported the median of the training losses at each learning step, along with the ranges between the 45th and 55th percentiles and between the 2.5th and 97.5th percentiles.

**Results.** As shown in Figure 1, we present the training losses of $\alpha$-Div across the learning steps for $\alpha = -2.0, 3.0$, and $0.5$. Results for other values of $\alpha$ are provided in Section D.1 in the Appendix. The figures on the left ($\alpha = -2.0$) and in the center ($\alpha = 3.0$) show that the training losses diverged to negative infinity when $\alpha < 0$ or $\alpha > 1$. In contrast, the figure on the right ($\alpha = 0.5$) demonstrates that the training losses successfully converged. These results highlight the stability of $\alpha$-Div's optimization when $\alpha$ is within the interval $(0, 1)$, as discussed in Section 5.1.

### 6.2 EXPERIMENTS ON THE IMPROVEMENT OF OPTIMIZATION EFFICIENCY BY REMOVING GRADIENT BIAS

Unbiased gradients of loss functions are expected to enhance the optimization of neural network parameters by ensuring that updates are made in directions that are closer to the ideal ones at each iteration, compared to biased gradient loss functions. We empirically compared the efficiency of minimizing training losses between the proposed loss function and the standard $\alpha$-divergence loss function derived from Equation (12). Our observations indicated that the proposed loss function was more effective in minimizing training loss than the biased-gradient $\alpha$-divergence loss function. Additionally, we found that the estimated density ratios using the standard $\alpha$-divergence loss function remained close to zero, whereas those obtained using $\alpha$-Div did not. This finding is consistent with the discussion in Section 5.3.

**Experimental Setup.** We first generated 100 training datasets from two normal distributions, $P = \mathcal{N}(\mu_p, I_5)$ and $Q = \mathcal{N}(\mu_q, I_5)$, where $I_5$ denotes the 5-dimensional identity matrix. The means were set as $\mu_p = (-5/2, 0, 0, 0, 0)$ and $\mu_q = (5/2, 0, 0, 0, 0)$. We then trained neural networks using three different loss functions: the standard $\alpha$-divergence loss function defined in Equation (12), $\alpha$-Div, and deep direct DRE (D3RE) (Kato & Teshima, 2021). Training losses were measured at each learning step. D3RE is designed to prevent issues such as train-hacking associated with Bregman divergence loss functions, as described in Section 3.4, by addressing the lower-unboundedness of the loss functions. The hyperparameter for D3RE was set to $C = 2$. We chose the Bregman divergence least-squares importance fitting (nnBD-LSIF) loss function for D3RE due to its unbiased gradient and expected stable optimization. For both the standard $\alpha$-divergence loss and $\alpha$-Div, we used

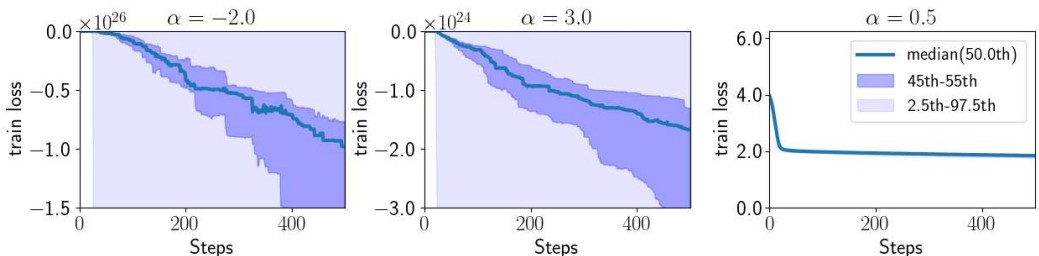

Figure 1: Results from Section 6.1. The left ($\alpha = -2.0$), center ($\alpha = 3.0$), and right ($\alpha = 0.5$) graphs show the training losses (y-axis) over learning steps (x-axis) during optimization using $\alpha$-Div for different $\alpha$ values. The solid blue line represents the median training losses, the dark blue shaded area shows the 45th to 55th percentiles, and the light blue shaded area represents the 2.5th to 97.5th percentiles.

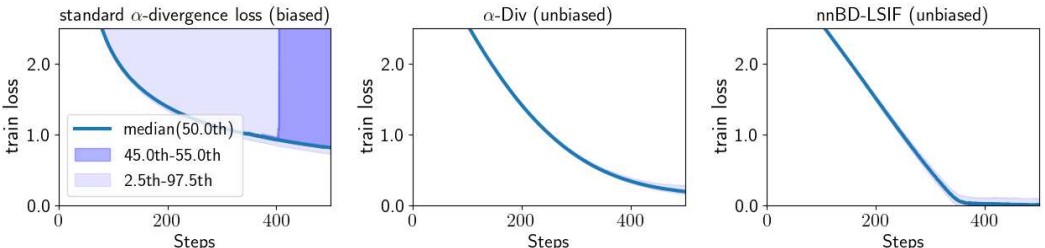

Figure 2: Results from Section 6.2. The $y$-axis of the left, center, and right graphs represents training losses during optimization using $\alpha$-Div, the biased-gradient $\alpha$-divergence loss function, and nnBD-LSIF, respectively. The $x$-axis represents learning steps. The solid blue line indicates the median training losses, the dark blue shaded areas show the 45th to 55th percentiles, and the light blue shaded areas show the 2.5th to 97.5th percentiles.

$\alpha = 0.5$. Finally, we reported the median training losses at each learning step, along with ranges between the 45th and 55th percentiles and between the 2.5th and 97.5th percentiles.

**Results.** Figure 2 illustrates the training losses at each learning step for each loss function. The center and right panels show that $\alpha$-Div and nnBD-LSIF are more effective at minimizing training losses compared to the standard $\alpha$-divergence loss function. These findings indicate that the unbiased gradient of $\alpha$-Div, like nnBD-LSIF, leads to more efficient neural network optimization than the biased gradient of the standard $\alpha$-divergence loss function. Additionally, while the training losses for the standard $\alpha$-divergence loss function diverged after 400 steps, those for $\alpha$-Div remained stable. According to Equation (12), the estimated density ratio $\phi_\theta$ in Equation (12) approaches 0 or $\infty$ as $\mathcal{L}_{\alpha\text{-standard}}(\phi_\theta) \to \infty$. However, $\phi_\theta \to \infty$ does not occur because there is neither a stable point where $\mathcal{L}_{\alpha\text{-standard}}(\phi_\theta) \to c$ for some constant $c$ nor does $\mathcal{L}_{\alpha\text{-standard}}(\phi_\theta)$ decrease monotonically to an extremely negative value. Therefore, it is considered that $\phi_\theta \to 0$ when $\mathcal{L}_{\alpha\text{-standard}}(\phi_\theta) \to \infty$, which causes the gradients of $\mathcal{L}_{\alpha\text{-standard}}(\phi_\theta)$ to vanish, i.e., $\nabla_\theta \mathcal{L}_{\alpha\text{-standard}}(\phi_\theta) \to 0$, as shown for $0 < \alpha < 1$ in Table 4. In contrast, this issue was not observed for $\alpha$-Div, consistent with the discussion in Section 5.3.

### 6.3 Experiments on the Estimation Accuracy Using High KL-Divergence Data

In Section 3.5, we examined the $\alpha$-divergence loss function, hypothesizing that its boundedness could address sample size issues in high KL-divergence data. Theorem 5.2, based on this boundedness, suggests that $\alpha$-Div can be minimized regardless of the true KL-divergence, indicating its potential for effective DRE with high KL-divergence data. To validate this hypothesis, we assessed DRE and KL-divergence estimation accuracy using both $\alpha$-Div and a KL-divergence loss function.

Unfortunately, the RMSE of DRE using $\alpha$-Div increased significantly with higher KL-divergence in the test datasets, similar to the results obtained with the KL-divergence loss function. Additionally,

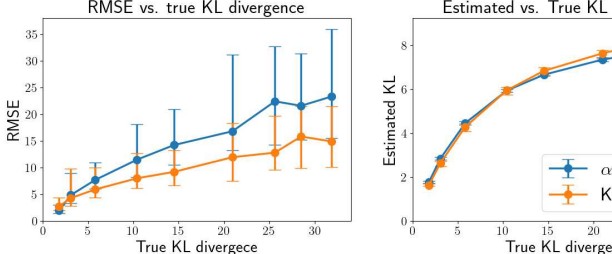

Figure 3: Results of Section 6.3. The $x$-axis represents the ground truth KL-divergence of the data. The $y$-axis of the left and right graphs represents the RMSE and estimated KL-divergence, respectively. The plot shows the median $y$-axis values for the ground truth KL-divergence. Vertical lines indicate the interquartile range (25th to 75th percentiles) of the $y$-axis values.

both methods produced nearly identical KL-divergence estimations. These findings suggest that the accuracy of DRE and KL-divergence estimation is primarily influenced by the true amount of KL-divergence in the data rather than the $\alpha$-divergence.

**Experimental Setup.** We generated 100 training and 100 test datasets, each containing 10,000 samples. The datasets were drawn from two normal distributions, $P = \mathcal{N}(\mu_p, \sigma^2 \cdot I_3)$ and $Q = \mathcal{N}(\mu_q, 4^2 \cdot I_3)$, where $\mu_p = (-3/2, -3/2, -3/2)$ and $\mu_q = (3/2, 3/2, 3/2)$, with $I_3$ denoting the 3-dimensional identity matrix. The values of $\sigma$ were set to 1.0, 1.1, 1.2, 1.4, 1.6, 2.0, 2.5, and 3.0. Correspondingly, the ground truth KL-divergence values of the datasets were 31.8, 25.6, 21.0, 14.5, 10.4, 5.8, 3.1, and 1.8 nats[1], reflecting the increasing $\sigma^2$ values. The true density ratios of the test datasets are known for this experimental setup. We trained neural networks on the training datasets by optimizing both $\alpha$-Div with $\alpha = 0.5$ and the KL-divergence loss function. After training, we measured the root mean squared error (RMSE) of the estimated density ratios using the test datasets. Additionally, we estimated the KL-divergence of the test datasets based on the estimated density ratios using a plug-in estimator. Finally, we reported the median RMSE of the DRE and the estimated KL-divergence, along with the interquartile range (25th to 75th percentiles), for both the KL-divergence loss function and $\alpha$-Div.

**Results.** Figure 3 shows the experimental results. The x-axis represents the true KL-divergence values of the test datasets, while the y-axes of the graphs display the RMSE (left) and estimated KL-divergence (right) for the test datasets. We empirically observed that the RMSE for DRE using $\alpha$-Div increased significantly as the KL-divergence of the datasets increased. A similar trend was observed for the KL-divergence loss function. Additionally, the KL-divergence estimation results were nearly identical for both methods. These findings indicate that the accuracy of DRE and KL-divergence estimation is primarily determined by the amount of KL-divergence in the data and is less influenced by $\alpha$-divergence. Therefore, we conclude that the approach discussed in Section 5.4 offers no advantage over the KL-divergence loss function in terms of the RMSE for DRE with high KL-divergence data. However, we believe that these empirical findings contribute to a deeper understanding of the accuracy of downstream tasks in DRE using $f$-divergence loss functions.

## 7    CONCLUSION

This study introduced a novel loss function for DRE, $\alpha$-Div, which is both concise and provides stable, efficient optimization. We offered technical justifications and demonstrated its effectiveness through numerical experiments. The empirical results affirmed the efficiency of the proposed loss function. However, experiments with high KL-divergence data revealed that the $\alpha$-divergence loss function does not offer a significant advantage over the KL-divergence loss function in terms of RMSE for DRE. These findings contribute to a deeper understanding of the accuracy of downstream tasks in DRE when using $f$-divergence loss functions.

---

[1]A 'nat' is a unit of information measured using the natural logarithm (base $e$)

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

## A    ORGANIZATION OF THE SUPPLEMENTARY DOCUMENT

The organization of this supplementary document is as follows: Section B reviews prior work in DRE using $f$-divergence optimization. Section C presents the theorems and proofs cited in this study. Section D provides details of the numerical experiments conducted. Finally, Section D presents additional experimental results. Additionally, the code used in the numerical experiments is included as supplementary material.

## B    RELATED WORK

Nguyen et al. (2010) proposed DRE using variational representations of $f$-divergences. Sugiyama et al. (2012) introduced the density-ratio matching under the Bregman divergence, which is a general framework that unifies various methods for DRE. As Sugiyama et al. (2012) mentioned, the density-ratio matching under the Bregman divergence is equivalent to DRE using variational representations of $f$-divergences. For estimation with high KL-divergence data, Rhodes et al. (2020) proposed a method that divides the high KL-divergence estimation into multiple smaller divergence estimations. Choi et al. (2022) further developed a continuous decomposition approach by introducing an auxiliary variable for transforming the data distribution. DRE using variational representations of $f$-divergences has also been studied from the perspective of classification-based modeling. Menon & Ong (2016) demonstrated that DRE through $f$-divergence optimization can be represented as a binary classification problem. Kato et al. (2019) proposed using the risk functions in PU learning for DRE.

Finally we review prior studies on DRE regarding $\alpha$-divergence loss functions. Birrell et al. (2021) derived an $\alpha$-divergence loss function from R'enyi's $\alpha$-divergence, while Cai et al. (2020) used a standard variational representation of Amari's $\alpha$-divergence with $\alpha < 0$ or $\alpha > 1$. Kwon & Baek (2024) presented the same $\alpha$-divergence loss function as proposed in this study, using the Gibbs density expression to measure entropy in thermodynamics. In contrast, we offer the proposed loss function to address the biased gradient problem.

## C    PROOFS

In this section, we present the theorems and proofs referenced in this study. First, we define $\alpha$-Div within a probabilistic theoretical framework. Following that, we provide the theorems and proofs cited throughout the study.

**Capital, small and bold letters.**    Random variables are denoted by capital letters. For example, $X$. Small letters are used for values of the random variables of the corresponding capital letters; $a$ denotes a value of the random variable $X$. Bold letters $\mathbf{X}$ and $\mathbf{x}$ represent sets of random variables and their values.

### C.1    DEFINITION OF $\alpha$-DIV

**Definition C.1** ($\alpha$-Divergence loss). Let $\mathbf{X}_P^1, \mathbf{X}_P^2, \ldots, \mathbf{X}_P^R$ denote $R$ i.i.d. random variables drawn from $P$, and let $\mathbf{X}_Q^1, \mathbf{X}_Q^2, \ldots, \mathbf{X}_Q^S$ denote $S$ i.i.d. random variables drawn from $Q$. Then, the $\alpha$-Divergence loss $\mathcal{L}_{\alpha\text{-Div}}^{(R,S)}(\,\cdot\,;\,\alpha)$ is defined as follows:

$$\mathcal{L}_{\alpha\text{-Div}}^{(R,S)}(T\,;\,\alpha) = \frac{1}{\alpha} \cdot \frac{1}{S} \cdot \sum_{i=1}^{S} e^{\alpha \cdot T(\mathbf{X}_Q^i)} + \frac{1}{1-\alpha} \cdot \frac{1}{R} \cdot \sum_{i=1}^{R} e^{(\alpha-1) \cdot T(\mathbf{X}_P^i)}, \tag{17}$$

where $T$ is a measurable function over $\Omega$ such that $T : \Omega \to \mathbb{R}$.

### C.2    PROOFS FOR SECTION 4

In this section, we provide the theorems and proofs referenced in Section 4.

**Theorem C.2.** *A variational representation of $\alpha$-divergence is given as*

$$D(Q\|P\,;\,\alpha) = \sup_{\phi \geq 0} \left\{ \frac{1}{\alpha \cdot (1-\alpha)} - \frac{1}{\alpha} \cdot E_Q\left[\phi^{-\alpha}\right] - \frac{1}{1-\alpha} \cdot E_P\left[\phi^{1-\alpha}\right] \right\}, \tag{18}$$

*where the supremum is taken over all measurable functions with $E_P[\phi^{1-\alpha}] < \infty$ and $E_Q[\phi^{-\alpha}] < \infty$. The maximum value is achieved at $\phi = dQ/dP$.*

*Proof of Theorem C.2.* Let $f_\alpha(t) = \{t^{1-\alpha} - (1-\alpha) \cdot t - \alpha\}/\{\alpha \cdot (\alpha - 1)\}$ for $\alpha \neq 0, 1$, then

$$E_P\left[f_\alpha\left(\frac{dQ}{dP}\right)\right] = E_P\left[\frac{1}{\alpha \cdot (1-\alpha)} \cdot \left(\frac{dQ}{dP}\right)^{1-\alpha} + \frac{1}{\alpha} \cdot \left(\frac{dQ}{dP}\right) + \frac{1}{1-\alpha}\right]$$

$$= \frac{1}{\alpha \cdot (1-\alpha)} \cdot E_P\left[\left(\frac{dQ}{dP}\right)^{1-\alpha}\right] + \frac{1}{\alpha} + \frac{1}{1-\alpha}$$

$$= D(Q\|P\,;\,\alpha). \tag{19}$$

Note that, the Legendre transform of $g_\alpha(x) = x^{1-\alpha}/(1-\alpha)$ is obtained as

$$g_\alpha^*(x) = \frac{\alpha}{\alpha - 1} \cdot x^{1 - \frac{1}{\alpha}}, \tag{20}$$

and for the Legendre transforms of functions, it holds that

$$\{C \cdot h(x)\}^* = C \cdot h^*\left(\frac{x}{C}\right) \quad \text{and} \quad \{h(x) + C \cdot x + D\}^* = h^*(x - C) - D. \tag{21}$$

Here, $A^*$ denotes the Legendre transform of $A$.

From Equations (20) and (21), we have

$$f_\alpha^*(t) = \left\{ \frac{1}{(-\alpha)} \cdot g_\alpha(t) + \frac{1}{\alpha} \cdot t + \frac{1}{1-\alpha} \right\}^*$$

$$= \frac{1}{(-\alpha)} \cdot g_\alpha^*\left(-\alpha \cdot \left\{t - \frac{1}{\alpha}\right\}\right) - \frac{1}{1-\alpha}$$

$$= -\frac{1}{\alpha} \cdot g_\alpha^*(1 - \alpha t) + \frac{1}{\alpha - 1}$$

$$= -\frac{1}{\alpha} \cdot \left\{ \frac{\alpha}{\alpha - 1} \cdot (1 - \alpha t)^{1 - \frac{1}{\alpha}} \right\} + \frac{1}{\alpha - 1}$$

$$= \frac{1}{1-\alpha} \cdot (1 - \alpha t)^{1 - \frac{1}{\alpha}} + \frac{1}{\alpha - 1}. \tag{22}$$

By differentiating $f_\alpha(t)$, we obtain

$$f_\alpha'(t) = -\frac{1}{\alpha} \cdot t^{-\alpha} + \frac{1}{\alpha}. \tag{23}$$

Thus,

$$E_Q\left[f_\alpha'(\phi)\right] = E_Q\left[-\frac{1}{\alpha} \cdot \phi^{-\alpha} + \frac{1}{\alpha}\right]. \tag{24}$$

From (22) and (23), we have

$$E_P\left[f_\alpha^*(f_\alpha'(\phi))\right] = E_P\left[\frac{1}{1-\alpha} \cdot \left\{1 - \alpha \cdot \left(-\frac{1}{\alpha} \cdot \phi^{-\alpha} + \frac{1}{\alpha}\right)\right\}^{1 - \frac{1}{\alpha}} + \frac{1}{\alpha - 1}\right]$$

$$= E_P\left[\frac{1}{1-\alpha} \cdot \phi^{1-\alpha} + \frac{1}{\alpha - 1}\right]. \tag{25}$$

In addition, from Equations (24) and (25), we observe that $E_P\left[\phi^{1-\alpha}\right] < \infty$ is equivalent to $E_P\left[\left|f_\alpha^*(f_\alpha'(\phi))\right|\right] < \infty$. Similarly, $E_Q\left[\phi^{-\alpha}\right] < \infty$ is equivalent to $E_Q\left[\left|f_\alpha'(\phi)\right|\right] < \infty$.

Finally, by substituting Equations (24) and (25) into Equation (1), we get

$$D(Q||P\,;\,\alpha) = \sup_{\phi \geq 0} \left\{ E_Q\left[f'_\alpha(\phi)\right] - E_P\left[f^*_\alpha(f'_\alpha(\phi))\right] \right\}$$

$$= \sup_{\phi \geq 0} \left\{ E_Q\left[ -\frac{1}{\alpha} \cdot \phi^{-\alpha} + \frac{1}{\alpha} \right] - E_P\left[ \frac{1}{1-\alpha} \cdot \phi^{1-\alpha} + \frac{1}{\alpha-1} \right] \right\}$$

$$= \sup_{\phi \geq 0} \left\{ \frac{1}{\alpha \cdot (1-\alpha)} - \frac{1}{\alpha} \cdot E_Q\left[\phi^{-\alpha}\right] - \frac{1}{1-\alpha} \cdot E_P\left[\phi^{1-\alpha}\right] \right\}.$$

This completes the proof. □

**Theorem C.3** (Theorem 4.2 in Section 4 restated). *The $\alpha$-divergence is represented as*

$$D(Q||P\,;\,\alpha) = \sup_{T:\Omega \to \mathbb{R}} \left\{ \frac{1}{\alpha \cdot (1-\alpha)} - \frac{1}{\alpha} \cdot E_Q\left[e^{\alpha \cdot T}\right] - \frac{1}{1-\alpha} \cdot E_P\left[e^{(\alpha-1) \cdot T}\right] \right\}, \quad (26)$$

*where the supremum is taken over all measurable functions $T : \Omega \to \mathbb{R}$ with $E_P[e^{(\alpha-1) \cdot T}] < \infty$ and $E_Q[e^{\alpha \cdot T}] < \infty$. The equality holds for $T^*$ satisfying*

$$\frac{dQ}{dP} = e^{-T^*}. \quad (27)$$

*proof of Theorem 4.2.* Substituting $e^{-T}$ into $\phi$ in Equation (18), we have

$$D(Q||P\,;\,\alpha) = \sup_{\phi \geq 0} \left\{ \frac{1}{\alpha \cdot (1-\alpha)} - \frac{1}{\alpha} \cdot E_Q\left[\phi^{-\alpha}\right] - \frac{1}{1-\alpha} \cdot E_P\left[\phi^{1-\alpha}\right] \right\}$$

$$= \sup_{T:\Omega \to \mathbb{R}} \left\{ \frac{1}{\alpha \cdot (1-\alpha)} - \frac{1}{\alpha} \cdot E_Q\left[\{e^{-T}\}^{-\alpha}\right] - \frac{1}{1-\alpha} \cdot E_P\left[\{e^{-T}\}^{1-\alpha}\right] \right\}$$

$$= \sup_{T:\Omega \to \mathbb{R}} \left\{ \frac{1}{\alpha \cdot (1-\alpha)} - \frac{1}{\alpha} \cdot E_Q\left[e^{\alpha \cdot T}\right] - \frac{1}{1-\alpha} \cdot E_P\left[e^{(\alpha-1) \cdot T}\right] \right\}. \quad (28)$$

Finally, from Theorem C.2, the equality for Equation (28) holds if and only if

$$\frac{dQ}{dP} = e^{-T^*}. \quad (29)$$

This completes the proof. □

**Lemma C.4.** *For a measurable function $T : \Omega \to \mathbb{R}$ with $E_P[e^{(\alpha-1) \cdot T}] < \infty$ and $E_Q[e^{\alpha \cdot T}] < \infty$, let*

$$\widetilde{l}_{\alpha\text{-}Div}\left(T(\mathbf{x})\,;\,\alpha\right) = \frac{1}{\alpha} \cdot e^{\alpha \cdot T(\mathbf{x})} \cdot \frac{dQ}{d\mu}(\mathbf{x}) + \frac{1}{1-\alpha} \cdot e^{(\alpha-1) \cdot T(\mathbf{x})} \cdot \frac{dP}{d\mu}(\mathbf{x}). \quad (30)$$

*Then the optimal function $T^*$ for $\inf_{T:\Omega \to \mathbb{R}} \widetilde{l}_{\alpha\text{-}Div}\left(T\,;\,\alpha\right)$ is obtained as $T^* = -\log dQ/dP$, $\mu$-almost everywhere.*

*proof of Lemma C.4.* First, note that it follows from Jensen's inequality that

$$\log(p \cdot X + q \cdot Y) \geq p \cdot \log(X) + q \cdot \log(Y), \quad (31)$$

for $X, Y > 0$ and $p, q > 0$ with $p + q = 1$, and equality holds when $X = Y$.

Substitute $X = e^{\alpha \cdot T(\mathbf{x})} \cdot \frac{dQ}{d\mu}(\mathbf{x})$, $Y = e^{(\alpha-1) \cdot T(\mathbf{x})} \cdot \frac{dP}{d\mu}(\mathbf{x})$, $p = 1 - \alpha$, and $q = \alpha$ into Equation (31), we obtain

$$\log(p \cdot X + q \cdot Y) = \log\left( \frac{1}{\alpha \cdot (1-\alpha)} \cdot \widetilde{l}_{\alpha\text{-}Div}(T\,;\,\alpha) \right),$$

and $\log\left( \frac{1}{\alpha \cdot (1-\alpha)} \cdot \widetilde{l}_{\alpha\text{-}Div}(T\,;\,\alpha) \right)$ is minimized when $e^{\alpha \cdot T(\mathbf{x})} \cdot \frac{dQ}{d\mu}(\mathbf{x}) = e^{(\alpha-1) \cdot T(\mathbf{x})} \cdot \frac{dP}{d\mu}(\mathbf{x})$, $\mu$-almost everywhere. Therefore, $\inf_{T:\Omega \to \mathbb{R}} \widetilde{l}_{\alpha\text{-}Div}(T\,;\,\alpha)$ is achieved at $e^{-T^*(\mathbf{x})} = \frac{dQ}{dP}(\mathbf{x})$, $\mu$-almost everywhere. Thus, we obtain $T^*(\mathbf{x}) = -\log \frac{dQ}{dP}(\mathbf{x})$, $\mu$-almost everywhere.

This completes the proof. □

**Lemma C.5.** *For a measurable function $T : \Omega \to \mathbb{R}$ with $E_P[e^{(\alpha-1)\cdot T}] < \infty$ and $E_Q[e^{\alpha\cdot T}] < \infty$, let*

$$\overline{\mathcal{L}}_{\alpha\text{-}Div}(T \, ; \, \alpha) = E_\mu \left[ \widetilde{l}_{\alpha\text{-}Div}\left(T(\mathbf{x} \, ; \, \alpha)\right) \right] \tag{32}$$

$$= \frac{1}{\alpha} \cdot E_Q \left[ e^{\alpha \cdot T(\mathbf{x})} \right] + \frac{1}{1-\alpha} \cdot E_P \left[ e^{(\alpha-1)\cdot T(\mathbf{x})} \right], \tag{33}$$

*and let*

$$\widetilde{l}^*_{\alpha\text{-}Div}(\alpha) = \inf_{T:\Omega\to\mathbb{R}} \widetilde{l}_{\alpha\text{-}Div}(T \, ; \, \alpha), \quad \text{and} \tag{34}$$

$$\overline{\mathcal{L}}^*_{\alpha\text{-}Div}(\alpha) = \inf_{T:\Omega\to\mathbb{R}} \overline{\mathcal{L}}_{\alpha\text{-}Div}(T \, ; \, \alpha), \tag{35}$$

*where the infima of Equations (34) and (35) are considered over measurable functions $T : \Omega \to \mathbb{R}$ with $E_P[e^{(\alpha-1)\cdot T}] < \infty$ and $E_Q[e^{\alpha\cdot T}] < \infty$.*

*Then,*

$$E_\mu \left[ \widetilde{l}^*_{\alpha\text{-}Div}(\alpha) \right] = \overline{\mathcal{L}}^*_{\alpha\text{-}Div}(\alpha). \tag{36}$$

*Additionally, the equality in Equations (34) and (35) hold for $T^*(\mathbf{x}) = -\log dQ/dP(\mathbf{x})$.*

*proof of Lemma C.5.* Let $T^*(\mathbf{x}) = -\log dQ/dP(\mathbf{x})$.

First, it follows from Lemma C.4 that

$$\widetilde{l}^*_{\alpha\text{-Div}}(\alpha) = \inf_{T:\Omega\to\mathbb{R}} \widetilde{l}_{\alpha\text{-Div}}(T \, ; \, \alpha) = \widetilde{l}_{\alpha\text{-Div}}(T^* \, ; \, \alpha). \tag{37}$$

Next, we obtain

$$E_\mu \left[ \frac{1}{\alpha \cdot (1-\alpha)} - \widetilde{l}^*_{\alpha\text{-Div}}(\alpha) \right] = \int \left\{ \frac{1}{\alpha \cdot (1-\alpha)} - \widetilde{l}^*_{\alpha\text{-Div}}(\alpha) \right\} d\mu$$

$$= \int \left\{ \frac{1}{\alpha \cdot (1-\alpha)} - \inf_{T:\Omega\to\mathbb{R}} \widetilde{l}_{\alpha\text{-Div}}(T \, ; \, \alpha) \right\} d\mu$$

$$= \int \sup_{T:\Omega\to\mathbb{R}} \left\{ \frac{1}{\alpha \cdot (1-\alpha)} - \widetilde{l}_{\alpha\text{-Div}}(T \, ; \, \alpha) \right\} d\mu. \tag{38}$$

Let $T_k = T^* + 1/k$. Note that,

$$\lim_{k\to\infty} \widetilde{l}_{\alpha\text{-Div}}(T_k \, ; \, \alpha) = \inf_{T:\Omega\to\mathbb{R}} \widetilde{l}_{\alpha\text{-Div}}(T \, ; \, \alpha) = \widetilde{l}^*_{\alpha\text{-Div}}(\alpha). \tag{39}$$

Then, we have

$$\lim_{k\to\infty} \left\{ \frac{1}{\alpha \cdot (1-\alpha)} - \widetilde{l}_{\alpha\text{-Div}}(T_k \, ; \, \alpha) \right\}$$

$$= \frac{1}{\alpha \cdot (1-\alpha)} - \lim_{k\to\infty} \left\{ \widetilde{l}_{\alpha\text{-Div}}(T_k \, ; \, \alpha) \right\}$$

$$= \frac{1}{\alpha \cdot (1-\alpha)} - \inf_{T:\Omega\to\mathbb{R}} \widetilde{l}_{\alpha\text{-Div}}(T \, ; \, \alpha)$$

$$= \sup_{T:\Omega\to\mathbb{R}} \left\{ \frac{1}{\alpha \cdot (1-\alpha)} - \widetilde{l}_{\alpha\text{-Div}}(T \, ; \, \alpha) \right\}. \tag{40}$$

From Theorem C.3, we have

$$\lim_{k \to \infty} E_\mu \left[ \frac{1}{\alpha \cdot (1-\alpha)} - \widetilde{l}_{\alpha\text{-Div}}(T_k \, ; \, \alpha) \right]$$

$$= \frac{1}{\alpha \cdot (1-\alpha)} - \lim_{k \to \infty} E_\mu \left[ \widetilde{l}_{\alpha\text{-Div}}(T_k \, ; \, \alpha) \right]$$

$$= \frac{1}{\alpha \cdot (1-\alpha)} - \lim_{k \to \infty} \left\{ \frac{1}{\alpha} \cdot E_Q \left[ e^{\alpha \cdot T_k} \right] + \frac{1}{1-\alpha} \cdot E_P \left[ e^{(\alpha-1) \cdot T_k} \right] \right\}$$

$$= \frac{1}{\alpha \cdot (1-\alpha)} - \inf_{T:\Omega \to \mathbb{R}} \left\{ \frac{1}{\alpha} \cdot E_Q \left[ e^{\alpha \cdot T} \right] + \frac{1}{1-\alpha} \cdot E_P \left[ e^{(\alpha-1) \cdot T} \right] \right\}$$

$$= \frac{1}{\alpha \cdot (1-\alpha)} - \inf_{T:\Omega \to \mathbb{R}} E_\mu \left[ \widetilde{l}_{\alpha\text{-Div}}(T \, ; \, \alpha) \right]$$

$$= \sup_{T:\Omega \to \mathbb{R}} \left\{ \frac{1}{\alpha \cdot (1-\alpha)} - E_\mu \left[ \widetilde{l}_{\alpha\text{-Div}}(T \, ; \, \alpha) \right] \right\}$$

$$= \sup_{T:\Omega \to \mathbb{R}} E_\mu \left[ \frac{1}{\alpha \cdot (1-\alpha)} - \widetilde{l}_{\alpha\text{-Div}}(T \, ; \, \alpha) \right]. \tag{41}$$

Now, we have

$$\left| \frac{1}{\alpha \cdot (1-\alpha)} - \widetilde{l}_{\alpha\text{-Div}}(T_k \, ; \, \alpha) \right|$$

$$= \left| \frac{1}{\alpha \cdot (1-\alpha)} - \frac{1}{\alpha} \cdot \left( \frac{dQ}{dP}(\mathbf{x}) \right)^\alpha \cdot \frac{dQ}{d\mu}(\mathbf{x}) \cdot e^{\frac{\alpha}{k}} \right.$$

$$\left. - \frac{1}{1-\alpha} \cdot \left( \frac{dQ}{dP}(\mathbf{x}) \right)^{\alpha-1} \cdot \frac{dP}{d\mu}(\mathbf{x}) \cdot e^{\frac{\alpha-1}{k}} \right|$$

$$\leq \frac{1}{\alpha \cdot (1-\alpha)} + \frac{1}{\alpha} \cdot e^{\frac{\alpha}{k}} \cdot \left( \frac{dQ}{dP}(\mathbf{x}) \right)^\alpha \cdot \frac{dQ}{d\mu}(\mathbf{x})$$

$$+ \frac{1}{1-\alpha} \cdot e^{\frac{\alpha-1}{k}} \cdot \left( \frac{dQ}{dP}(\mathbf{x}) \right)^{\alpha-1} \cdot \frac{dP}{d\mu}(\mathbf{x}).$$

$$= \frac{1}{\alpha \cdot (1-\alpha)} + \left\{ \frac{1}{\alpha} \cdot e^{\frac{\alpha}{k}} + \frac{1}{1-\alpha} \cdot e^{\frac{\alpha-1}{k}} \right\} \cdot \left( \frac{dQ}{dP}(\mathbf{x}) \right)^{\alpha-1} \cdot \frac{dP}{d\mu}(\mathbf{x}), \tag{42}$$

and let $\phi(\mathbf{x})$ denote the term on the right hand side of Equation (42).

Then, we observe that

$$\left| \frac{1}{\alpha \cdot (1-\alpha)} - \widetilde{l}_{\alpha\text{-Div}}(T_k(\mathbf{x}) \, ; \, \alpha) \right| \leq \phi(\mathbf{x}) \quad \text{and} \quad E_\mu \left[ \phi(\mathbf{x}) \right] < \infty.$$

That is, the following sequence is uniformly integrable for $\mu$:

$$\left\{ \frac{1}{\alpha \cdot (1-\alpha)} - \widetilde{l}_{\alpha\text{-Div}}(T_k \, ; \, \alpha) \right\}_{k=1}^{\infty}.$$

Thus, from the property of the Lebesgue integral (Shiryaev, P188, Theorem 4), we have

$$E_\mu \left[ \lim_{k \to \infty} \left\{ \frac{1}{\alpha \cdot (1-\alpha)} - \widetilde{l}_{\alpha\text{-Div}}(T_k \, ; \, \alpha) \right\} \right] = \lim_{k \to \infty} E_\mu \left[ \frac{1}{\alpha \cdot (1-\alpha)} - \widetilde{l}_{\alpha\text{-Div}}(T_k \, ; \, \alpha) \right]. \tag{43}$$

From Equations (40), (41) and (43), we obtain

$$\frac{1}{\alpha \cdot (1-\alpha)} - E_\mu \left[ \widetilde{l}^*_{\alpha\text{-Div}}(\alpha) \right] = E_\mu \left[ \frac{1}{\alpha \cdot (1-\alpha)} - \widetilde{l}^*_{\alpha\text{-Div}}(\alpha) \right]$$

$$= E_\mu \left[ \sup_{T:\Omega \to \mathbb{R}} \left\{ \frac{1}{\alpha \cdot (1-\alpha)} - \widetilde{l}_{\alpha\text{-Div}}(T\,;\,\alpha) \right\} \right]$$

$$= E_\mu \left[ \lim_{k \to \infty} \left\{ \frac{1}{\alpha \cdot (1-\alpha)} - \widetilde{l}_{\alpha\text{-Div}}(T_k\,;\,\alpha) \right\} \right]$$

$$(\because \text{Equation (40)})$$

$$= \lim_{k \to \infty} E_\mu \left[ \frac{1}{\alpha \cdot (1-\alpha)} - \widetilde{l}_{\alpha\text{-Div}}(T_k\,;\,\alpha) \right] \qquad (\because \text{Equation (43)})$$

$$= \sup_{T:\Omega \to \mathbb{R}} \left\{ E_\mu \left[ \frac{1}{\alpha \cdot (1-\alpha)} - \widetilde{l}_{\alpha\text{-Div}}(T_k\,;\,\alpha) \right] \right\}$$

$$(\because \text{Equation (41)})$$

$$= \frac{1}{\alpha \cdot (1-\alpha)} - \inf_{T:\Omega \to \mathbb{R}} E_\mu \left[ \widetilde{l}_{\alpha\text{-Div}}(T_k\,;\,\alpha) \right]$$

$$= \frac{1}{\alpha \cdot (1-\alpha)} - \overline{\mathcal{L}}^*_{\alpha\text{-Div}}(\alpha).$$

Here, we have

$$E_\mu \left[ \widetilde{l}^*_{\alpha\text{-Div}}(\alpha) \right] = \overline{\mathcal{L}}^*_{\alpha\text{-Div}}(\alpha). \tag{44}$$

From Equations (32) and (44, we have

$$\overline{\mathcal{L}}_{\alpha\text{-Div}}(T^*\,;\,\alpha) = E_\mu \left[ \widetilde{l}^*_{\alpha\text{-Div}}(\alpha) \right] = \overline{\mathcal{L}}^*_{\alpha\text{-Div}}(\alpha). \tag{45}$$

This completes the proof. $\qquad\square$

**Theorem C.6** (Theorem 4.4 in Section 4 restated). *For a fixed function $T : \Omega \to \mathbb{R}$, let $c_*$ be the optimal scalar value for the following infimum:*

$$c_* = \arg\inf_{c \in \mathbb{R}} E[\mathcal{L}_{\alpha\text{-Div}}(T + c\,;\,\alpha)]$$

$$= \arg\inf_{c \in \mathbb{R}} \left\{ \frac{1}{\alpha} E_Q \left[ e^{\alpha \cdot (T+c)} \right] \right.$$

$$\left. + \quad \frac{1}{1-\alpha} E_P \left[ e^{(\alpha-1) \cdot (T+c)} \right] \right\}, \tag{46}$$

*Then, $c_*$ satisfies $e^{c_*} = E_P \left[ e^{-T} \right]$, or equivalently, $e^{-(T+c_*)} = e^{-T}/E_P \left[ e^{-T} \right]$.*

*proof of Theorem C.6.* Now, we have

$$\widetilde{l}_{\alpha\text{-Div}}(T + c\,;\,\alpha) = \frac{1}{\alpha} \cdot e^{\alpha \cdot c} \cdot e^{\alpha \cdot T(\mathbf{x})} \cdot \frac{dQ}{d\mu}(\mathbf{x}) + \frac{1}{1-\alpha} \cdot e^{(\alpha-1) \cdot c} \cdot e^{(\alpha-1) \cdot T(\mathbf{x})} \cdot \frac{dP}{d\mu}(\mathbf{x}).$$

For Equation (31), let $X = e^{\alpha \cdot c} \cdot e^{\alpha \cdot T(\mathbf{x})} \cdot \frac{dQ}{d\mu}(\mathbf{x})$, $Y = e^{(\alpha-1) \cdot c} \cdot e^{(\alpha-1) \cdot T(\mathbf{x})} \cdot \frac{dP}{d\mu}(\mathbf{x})$, $p = 1 - \alpha$ and $q = \alpha$. Then, from Jensen's inequality, $\widetilde{l}_{\alpha\text{-Div}}(T + c\,;\,\alpha)$ is minimized at $c_*$ such taht $e^{\alpha \cdot c_*} \cdot e^{\alpha \cdot T(\mathbf{x})} \cdot \frac{dQ}{d\mu}(\mathbf{x}) = e^{(\alpha-1) \cdot c_*} \cdot e^{(\alpha-1) \cdot T(\mathbf{x})} \cdot \frac{dP}{d\mu}(\mathbf{x})$, $\mu$-almost everywhere.

Hence,

$$e^{c_*} \cdot \frac{dQ}{d\mu}(\mathbf{x}) = e^{-T(\mathbf{x})} \cdot \frac{dP}{d\mu}(\mathbf{x}).$$

By integrating both sides of the above equality over $\Omega$ with $\mu$, we obtain

$$e^{c_*} = E_P \left[ e^{-T} \right].$$

This completes the proof. $\qquad\square$

## C.3 PROOFS FOR SECTION 5

In this section, we provide the theorems and proofs referred to in Section 5.

**Theorem C.7** (Theorem 5.1 in Section 5 restated). *Let $T_\theta(\mathbf{x}) : \Omega \to \mathbb{R}$ be a function such that the map $\theta = (\theta_1, \theta_2, \ldots, \theta_p) \in \Theta \mapsto T_\theta(\mathbf{x})$ is differentiable for all $\theta$ and $\mu$-almost every $\mathbf{x} \in \Omega$. Assume for a point $\bar{\theta} \in \Theta$, it holds that $E_P[e^{(\alpha-1) \cdot T_{\bar{\theta}}}] < \infty$ and $E_Q[e^{\alpha \cdot T_{\bar{\theta}}}] < \infty$, and there exists a compact neighborhood of $\bar{\theta}$, denoted by $B_{\bar{\theta}}$, and a constant value $L$, such that $|T_\psi(\mathbf{x}) - T_{\bar{\theta}}(\mathbf{x})| < L\|\psi - \bar{\theta}\|$.*

*Then,*

$$E\left[\nabla_\theta \mathcal{L}_{\alpha\text{-}Div}^{(R,S)}(T\,;\,\alpha)\Big|_{\theta=\bar{\theta}}\right] = \nabla_\theta E\left[\mathcal{L}_{\alpha\text{-}Div}^{(R,S)}(T\,;\,\alpha)\right]\Big|_{\theta=\bar{\theta}}. \tag{47}$$

*proof of Theorem C.7.* We now consider the values, as $\psi \to \bar{\theta}$, of the following two integrals:

$$\int \frac{1}{\|\psi - \bar{\theta}\|} \left\{\frac{1}{\alpha} e^{\alpha \cdot T_\psi} - \frac{1}{\alpha} e^{\alpha \cdot T_{\bar{\theta}}}\right\} dQ, \tag{48}$$

and

$$\int \frac{1}{\|\psi - \bar{\theta}\|} \left\{\frac{1}{1-\alpha} e^{(\alpha-1) \cdot T_\psi} - \frac{1}{1-\alpha} e^{(\alpha-1) \cdot T_{\bar{\theta}}}\right\} dP. \tag{49}$$

Note that it follows from the intermediate value theorem that

$$\left|\frac{1}{\alpha} e^{\alpha \cdot x} - \frac{1}{\alpha} e^{\alpha \cdot y}\right| = |x - y| \cdot e^{\alpha \cdot \{y + \tau \cdot (x-y)\}} \quad (\exists \tau \in [0,1]\,). \tag{50}$$

By using the above equation with $x = T_\psi(\mathbf{x})$ and $y = T_{\bar{\theta}}(\mathbf{x})$ for the integrand of Equation (48), we have

$$\left|\frac{1}{\|\psi - \bar{\theta}\|} \cdot \left\{\frac{1}{\alpha} e^{\alpha \cdot T_\psi(\mathbf{x})} - \frac{1}{\alpha} e^{\alpha \cdot T_{\bar{\theta}}(\mathbf{x})}\right\}\right|$$

$$= \frac{1}{\|\psi - \bar{\theta}\|} \cdot \left|T_\psi(\mathbf{x}) - T_{\bar{\theta}}(\mathbf{x})\right| \cdot e^{\alpha \cdot \{T_{\bar{\theta}}(\mathbf{x}) + \tau_\mathbf{x} \cdot (T_\psi(\mathbf{x}) - T_{\bar{\theta}}(\mathbf{x}))\}} \qquad (\,\tau_\mathbf{x} \in [0,1]\,)$$

$$= \frac{1}{\|\psi - \bar{\theta}\|} \cdot \left|T_\psi(\mathbf{x}) - T_{\bar{\theta}}(\mathbf{x})\right| \cdot e^{\alpha \cdot \tau_\mathbf{x} \cdot (T_\psi(\mathbf{x}) - T_{\bar{\theta}}(\mathbf{x}))} \cdot e^{\alpha \cdot T_{\bar{\theta}}(\mathbf{x})}$$

$$\leq \frac{1}{\|\psi - \bar{\theta}\|} \cdot \left|T_\psi(\mathbf{x}) - T_{\bar{\theta}}(\mathbf{x})\right| \cdot e^{\alpha \tau_\mathbf{x} |T_\psi(\mathbf{x}) - T_{\bar{\theta}}(\mathbf{x})|} \cdot e^{\alpha \cdot T_{\bar{\theta}}(\mathbf{x})}$$

$$\leq L \cdot e^{\alpha \cdot L \cdot \|\psi - \bar{\theta}\|} \cdot e^{\alpha \cdot T_{\bar{\theta}}(\mathbf{x})}, \tag{51}$$

for all $\psi \in B_{\bar{\theta}}$.

Integrating the term on the left-hand side of Equation (51) with respect to $Q$, we have

$$\int \left|\frac{1}{\|\psi - \bar{\theta}\|} \cdot \left\{\frac{1}{\alpha} \cdot e^{\alpha \cdot T_\psi(\mathbf{x}^q)} - \frac{1}{\alpha} \cdot e^{\alpha \cdot T_{\bar{\theta}}(\mathbf{x}^q)}\right\}\right| dQ(\mathbf{x}^q)$$

$$\leq \int L \cdot e^{\alpha \cdot L \cdot \|\psi - \bar{\theta}\|} \cdot e^{\alpha \cdot T_{\bar{\theta}}(\mathbf{x}^q)} dQ(\mathbf{x}^q)$$

$$= L \cdot e^{\alpha \cdot L \cdot \|\psi - \bar{\theta}\|} \cdot E_Q\left[e^{\alpha \cdot T_{\bar{\theta}}}\right]. \tag{52}$$

Considering the supremum for $\psi \in B_{\bar{\theta}}$ in Equation (52), we obtain

$$\sup_{\psi \in B_{\bar{\theta}}} \left\{\int \left|\frac{1}{\|\psi - \bar{\theta}\|} \cdot \left\{\frac{1}{\alpha} \cdot e^{\alpha \cdot T_\psi} - \frac{1}{\alpha} e^{\alpha \cdot T_{\bar{\theta}}}\right\}\right| dQ\right\}$$

$$\leq \sup_{\psi \in B_{\bar{\theta}}} \left\{L \cdot e^{\alpha \cdot L \cdot \|\psi - \bar{\theta}\|} \cdot E_Q\left[e^{\alpha \cdot T_{\bar{\theta}}}\right]\right\}$$

$$= E_Q\left[e^{\alpha \cdot T_{\bar{\theta}}}\right] \cdot \sup_{\psi \in B_{\bar{\theta}}} L \cdot e^{\alpha \cdot L \cdot \|\psi - \bar{\theta}\|} < \infty, \tag{53}$$

since $B_{\bar{\theta}}$ is compact.

Therefore, the following set is uniformly integrable for $Q$:

$$\left\{ \frac{1}{\|\psi - \bar{\theta}\|} \left\{ \frac{1}{\alpha} \cdot e^{\alpha \cdot T_\psi(\mathbf{x}^q)} - \frac{1}{\alpha} e^{\alpha \cdot T_{\bar{\theta}}(\mathbf{x}^q)} \right\} \ : \ \psi \in B_{\bar{\theta}} \right\}. \tag{54}$$

Similarly, for Equation (49), we have

$$\sup_{\psi \in B_{\bar{\theta}}} \int \left| \frac{1}{\|\psi - \bar{\theta}\|} \cdot \left\{ \frac{1}{1-\alpha} e^{(\alpha-1) \cdot T_\psi(\mathbf{x}^p)} - \frac{1}{1-\alpha} e^{(\alpha-1) \cdot T_{\bar{\theta}}(\mathbf{x}^p)} \right\} \right| dP(\mathbf{x}^p)$$

$$\leq \sup_{\psi \in B_{\bar{\theta}}} \left\{ L \cdot e^{(1-\alpha)L \cdot \|\psi - \bar{\theta}\|} \cdot E_P \left[ e^{(1-\alpha) \cdot T_{\bar{\theta}}} \right] \right\}$$

$$= E_P \left[ e^{(1-\alpha) \cdot T_{\bar{\theta}}} \right] \cdot \sup_{\psi \in B_{\bar{\theta}}} L \cdot e^{(1-\alpha)L \cdot \|\psi - \bar{\theta}\|} < \infty. \tag{55}$$

Therefore, the following set is uniformly integrable for $P$:

$$\left\{ \frac{1}{\|\psi - \bar{\theta}\|} \cdot \left\{ \frac{1}{1-\alpha} \cdot e^{(\alpha-1) \cdot T_\psi(\mathbf{x}^p)} - \frac{1}{1-\alpha} \cdot e^{(\alpha-1) \cdot T_{\bar{\theta}}(\mathbf{x}^p)} \right\} \ : \ \psi \in B_{\bar{\theta}} \right\}. \tag{56}$$

Thus, the Lebesgue integral and $\lim_{\psi \to \bar{\theta}}$ are exchangeable for the set in Equation (56). Then, we have

$$\nabla_\theta E_Q \left[ \frac{1}{\alpha} \cdot e^{\alpha \cdot T_\theta(\mathbf{x}^q)} \right] \Bigg|_{\theta = \bar{\theta}}$$

$$= \lim_{\psi \to \bar{\theta}} \int \frac{1}{\|\psi - \bar{\theta}\|} \cdot \left\{ \frac{1}{\alpha} \cdot e^{\alpha \cdot T_\psi(\mathbf{x}^q)} - \frac{1}{\alpha} \cdot e^{\alpha \cdot T_{\bar{\theta}}(\mathbf{x}^q)} \right\} dQ(\mathbf{x}^q)$$

$$= \int \lim_{\psi \to \bar{\theta}} \left[ \frac{1}{\|\psi - \bar{\theta}\|} \cdot \left\{ \frac{1}{\alpha} \cdot e^{\alpha \cdot T_\psi(\mathbf{x}^q)} - \frac{1}{\alpha} \cdot e^{\alpha \cdot T_{\bar{\theta}}(\mathbf{x}^q)} \right\} \right] dQ(\mathbf{x}^q)$$

$$= E_Q \left[ \nabla_\theta \left( \frac{1}{\alpha} \cdot e^{\alpha \cdot T_\theta(\mathbf{x}^q)} \right) \Bigg|_{\theta = \bar{\theta}} \right]. \tag{57}$$

Similarly, we obtain

$$\nabla_\theta E_P \left[ \frac{1}{1-\alpha} \cdot e^{(\alpha-1) \cdot T_\theta(\mathbf{x}^p)} \right] \Bigg|_{\theta = \bar{\theta}}$$

$$= \lim_{\psi \to \bar{\theta}} \int \frac{1}{\|\psi - \bar{\theta}\|} \cdot \left\{ \frac{1}{1-\alpha} \cdot e^{(\alpha-1) \cdot T_\psi(\mathbf{x}^p)} - \frac{1}{1-\alpha} \cdot e^{(\alpha-1) \cdot T_{\bar{\theta}}(\mathbf{x}^p)} \right\} dP(\mathbf{x}^p)$$

$$= \int \lim_{\psi \to \bar{\theta}} \left[ \frac{1}{\|\psi - \bar{\theta}\|} \cdot \left\{ \frac{1}{1-\alpha} \cdot e^{(\alpha-1) \cdot T_\psi(\mathbf{x}^p)} - \frac{1}{1-\alpha} \cdot e^{(\alpha-1) \cdot T_{\bar{\theta}}(\mathbf{x}^p)} \right\} \right] dP(\mathbf{x}^p)$$

$$= E_P \left[ \nabla_\theta \left( \frac{1}{1-\alpha} \cdot e^{(\alpha-1) \cdot T_\theta(\mathbf{x}^p)} \right) \Bigg|_{\theta = \bar{\theta}} \right]. \tag{58}$$

From Equations (57) and (58), we have

$$E\left[\nabla_\theta \mathcal{L}_{\alpha\text{-Div}}^{(R,S)}(T_\theta\,;\,\alpha)\Big|_{\theta=\bar\theta}\right]$$

$$= E_P\left[E_Q\left[\nabla_\theta \mathcal{L}_{\alpha\text{-Div}}^{(R,S)}(T\,;\,\alpha)\Big|_{\theta=\bar\theta}\right]\right]$$

$$= E_P\left[E_Q\left[\nabla_\theta|_{\theta=\bar\theta}\left\{\frac{1}{\alpha}\cdot\frac{1}{S}\cdot\sum_{i=1}^{S}e^{\alpha\cdot T_\theta(\mathbf{x}_i^q)}+\frac{1}{1-\alpha}\cdot\frac{1}{R}\cdot\sum_{i=1}^{R}e^{(\alpha-1)\cdot T_\theta(\mathbf{x}_i^p)}\right\}\right]\right]$$

$$= E_P\left[E_Q\left[\frac{1}{\alpha}\cdot\frac{1}{S}\cdot\sum_{i=1}^{S}\nabla_\theta\left(e^{\alpha\cdot T_\theta(\mathbf{x}_i^p)}\right)\Big|_{\theta=\bar\theta}\right.\right.$$

$$\left.\left.+\frac{1}{1-\alpha}\cdot\frac{1}{R}\cdot\sum_{i=1}^{R}\nabla_\theta\left(e^{(\alpha-1)\cdot T_\theta(\mathbf{x}_i^p)}\right)\Big|_{\theta=\bar\theta}\right]\right]$$

$$= \frac{1}{\alpha}\cdot\frac{1}{S}\cdot\sum_{i=1}^{S}E_Q\left[\nabla_\theta\left(e^{\alpha\cdot T_\theta(\mathbf{x}_i^q)}\right)\right]\Big|_{\theta=\bar\theta}$$

$$+\frac{1}{1-\alpha}\cdot\frac{1}{R}\cdot\sum_{i=1}^{R}E_P\left[\nabla_\theta\left(e^{(\alpha-1)\cdot T_\theta(\mathbf{x}_i^p)}\right)\Big|_{\theta=\bar\theta}\right]$$

$$= \frac{1}{\alpha}\cdot\frac{1}{S}\cdot\sum_{i=1}^{S}\nabla_\theta E_Q\left[e^{\alpha\cdot T_\theta(\mathbf{x}_i^q)}\right]\Big|_{\theta=\bar\theta}$$

$$+\frac{1}{1-\alpha}\cdot\frac{1}{R}\cdot\sum_{i=1}^{R}\nabla_\theta E_P\left[e^{(\alpha-1)\cdot T_\theta(\mathbf{x}_i^p)}\right]\Big|_{\theta=\bar\theta}$$

$$= \nabla_\theta E_Q\left[\frac{1}{\alpha}\cdot\frac{1}{S}\cdot\sum_{i=1}^{S}e^{\alpha\cdot T_\theta(\mathbf{x}_i^q)}\right]\Big|_{\theta=\bar\theta}$$

$$+\nabla_\theta E_P\left[\frac{1}{1-\alpha}\cdot\frac{1}{R}\cdot\sum_{i=1}^{R}e^{(\alpha-1)\cdot T_\theta(\mathbf{x}_i^p)}\right]\Big|_{\theta=\bar\theta}$$

$$= \nabla_\theta\left\{E_P\left[E_Q\left[\frac{1}{\alpha}\cdot\frac{1}{S}\cdot\sum_{i=1}^{S}e^{\alpha\cdot T_\theta(\mathbf{x}_i^q)}\right.\right.\right.$$

$$\left.\left.\left.+\frac{1}{1-\alpha}\cdot\frac{1}{R}\cdot\sum_{i=1}^{R}e^{(\alpha-1)\cdot T_\theta(\mathbf{x}_i^p)}\right]\right]\right\}\Big|_{\theta=\bar\theta}$$

$$= \nabla_\theta E_P\left[E_Q\left[\mathcal{L}_{\alpha\text{-Div}}^{(R,S)}(T_\theta\,;\,\alpha)\right]\right]\Big|_{\theta=\bar\theta}$$

$$= \nabla_\theta E\left[\mathcal{L}_{\alpha\text{-Div}}^{(R,S)}(T_\theta\,;\,\alpha)\right]\Big|_{\theta=\bar\theta}. \tag{59}$$

This completes the proof. $\qquad\square$

**Theorem C.8** (Theorem 5.2 in Section 5 restated). *Assume $E_P\left[(dQ/dP(\mathbf{X}))^{2\cdot\alpha}\right]<\infty$. Let $T^* = -\log dQ/dP$. Subsequently, let*

$$\hat{D}^{(N)}(Q\|P\,;\,\alpha) = \frac{1}{\alpha\cdot(1-\alpha)} - \mathcal{L}_{\alpha\text{-Div}}^{(N,N)}(T^*\,;\,\alpha). \tag{60}$$

*Then, it holds that as $N\to\infty$,*

$$\sqrt{N}\left\{\hat{D}^{(N)}(Q\|P\,;\,\alpha) - D(Q\|P\,;\,\alpha)\right\} \xrightarrow{d} \mathcal{N}\left(0,\,\sigma_\alpha^2\right), \tag{61}$$

*where*

$$\sigma_\alpha^2 = C_\alpha^1\cdot D(Q\|P\,;\,2\alpha) + C_\alpha^2\cdot D(Q\|P\,;\,2\alpha-1)$$
$$+ C_\alpha^3\cdot D(Q\|P\,;\,\alpha)^2 + C_\alpha^4\cdot D(Q\|P\,;\,\alpha) + C_\alpha^5, \tag{62}$$

*and*

$$C_\alpha^1 = \frac{2\alpha \cdot (1 - 2\alpha)}{\alpha^2}, \tag{63}$$

$$C_\alpha^2 = \frac{2\alpha \cdot (1 - 2\alpha)}{(1 - \alpha)^2}, \tag{64}$$

$$C_\alpha^3 = -\frac{1}{\alpha^2} - \frac{1}{(1 - \alpha)^2}, \tag{65}$$

$$C_\alpha^4 = \frac{2}{\alpha^2} + \frac{2}{(1 - \alpha)^2}, $$

$$C_\alpha^5 = \left( \frac{1}{\alpha^2} + \frac{1}{(1 - \alpha)^2} \right) \cdot (2 - 2\alpha \cdot (1 - \alpha)). \tag{66}$$

*proof of Theorem C.8.* First, note that

$$\hat{D}^{(N)}(Q\|P\,;\alpha) = \frac{1}{\alpha \cdot (1 - \alpha)} - \frac{1}{\alpha} \cdot \left\{ \frac{1}{N} \cdot \sum_{i=1}^{N} e^{\alpha \cdot T^*(\mathbf{X}_Q^i)} \right\} - \frac{1}{1 - \alpha} \cdot \left\{ \frac{1}{N} \cdot \sum_{i=1}^{N} e^{(\alpha - 1) \cdot T^*(\mathbf{X}_P^i)} \right\}$$

$$= \frac{1}{\alpha \cdot (1 - \alpha)} - \frac{1}{\alpha} \cdot \left\{ \frac{1}{N} \cdot \sum_{i=1}^{N} \left( \frac{dQ}{dP}(\mathbf{X}_Q^i) \right)^{-\alpha} \right\}$$

$$- \frac{1}{1 - \alpha} \cdot \left\{ \frac{1}{N} \cdot \sum_{i=1}^{N} \left( \frac{dQ}{dP}(\mathbf{X}_P^i) \right)^{1-\alpha} \right\}. \tag{67}$$

On the other hand, from Lemma C.5, we obtain

$$D(Q\|P\,;\alpha) = \sup_{T:\Omega \to \mathbb{R}} \left\{ \frac{1}{\alpha \cdot (1 - \alpha)} - \frac{1}{\alpha} \cdot E_Q \left[ e^{\alpha \cdot T} \right] - \frac{1}{1 - \alpha} \cdot E_P \cdot \left[ e^{(\alpha - 1) \cdot T} \right] \right\}$$

$$= \frac{1}{\alpha \cdot (1 - \alpha)} - \inf_{T:\Omega \to \mathbb{R}} \left\{ \frac{1}{\alpha} \cdot E_Q \left[ e^{\alpha \cdot T} \right] - \frac{1}{1 - \alpha} \cdot E_P \left[ e^{(\alpha - 1) \cdot T} \right] \right\}$$

$$= \frac{1}{\alpha \cdot (1 - \alpha)} - \overline{\mathcal{L}}_{\alpha\text{-Div}}^*(\alpha)$$

$$= \frac{1}{\alpha \cdot (1 - \alpha)} - \frac{1}{\alpha} \cdot E_Q \left[ e^{\alpha \cdot T^*} \right] - \frac{1}{1 - \alpha} \cdot E_P \left[ e^{(\alpha - 1) \cdot T^*} \right]$$

$$= \frac{1}{\alpha \cdot (1 - \alpha)} - \frac{1}{\alpha} \cdot E_Q \left[ \left( \frac{dQ}{dP}(\mathbf{x}) \right)^{-\alpha} \right] - \frac{1}{1 - \alpha} \cdot E_P \left[ \left( \frac{dQ}{dP}(\mathbf{x}) \right)^{1-\alpha} \right]$$

$$= \frac{1}{\alpha \cdot (1 - \alpha)} - \frac{1}{\alpha} \cdot \left\{ \frac{1}{N} \cdot \sum_{i=1}^{N} E_Q \left[ \left( \frac{dQ}{dP}(\mathbf{x}_i) \right)^{-\alpha} \right] \right\}$$

$$- \frac{1}{1 - \alpha} \cdot \left\{ \frac{1}{N} \cdot \sum_{i=1}^{N} E_P \left[ \left( \frac{dQ}{dP}(\mathbf{x}_i) \right)^{1-\alpha} \right] \right\}. \tag{68}$$

Subtracting Equation (68) from Equation (67), we have

$$\hat{D}^{(N)}(Q||P\,;\,\alpha) - D(Q||P\,;\,\alpha)$$

$$= \frac{1}{N} \cdot \sum_{i=1}^{N} \frac{1}{\alpha} \cdot \left\{ \left( \frac{dQ}{dP}(\mathbf{X}_Q^i) \right)^{-\alpha} - E_Q \left[ \left( \frac{dQ}{dP}(\mathbf{x}_i) \right)^{-\alpha} \right] \right\}$$

$$+ \frac{1}{N} \cdot \sum_{i=1}^{N} \frac{1}{1-\alpha} \cdot \left\{ \left( \frac{dQ}{dP}(\mathbf{X}_P^i) \right)^{1-\alpha} - E_P \left[ \left( \frac{dQ}{dP}(\mathbf{x}_i) \right)^{1-\alpha} \right] \right\}$$

$$= \frac{1}{N} \cdot \sum_{i=1}^{N} \frac{1}{\alpha} \cdot \left\{ \left( \frac{dQ}{dP}(\mathbf{X}_Q^i) \right)^{-\alpha} - E_Q \left[ \left( \frac{dQ}{dP}(\mathbf{x}) \right)^{-\alpha} \right] \right\}$$

$$+ \frac{1}{N} \cdot \sum_{i=1}^{N} \frac{1}{1-\alpha} \cdot \left\{ \left( \frac{dQ}{dP}(\mathbf{X}_P^i) \right)^{1-\alpha} - E_P \left[ \left( \frac{dQ}{dP}(\mathbf{x}) \right)^{1-\alpha} \right] \right\}. \tag{69}$$

Let $L_Q^i = \frac{1}{\alpha} \cdot \left\{ \left( \frac{dQ}{dP}(\mathbf{X}_Q^i) \right)^{-\alpha} - E_Q \left[ \left( \frac{dQ}{dP}(\mathbf{x}) \right)^{-\alpha} \right] \right\}$. Then $\{L_Q^i\}_{i=1}^N$ are independent and identically distributed variables whose means and variances are as follows:

$$E_Q \left[ L_Q^i \right] = 0, \tag{70}$$

and

$$\mathrm{Var}_Q \left[ L_Q^i \right]$$

$$= E_Q \left[ \frac{1}{\alpha^2} \cdot \left\{ \left( \frac{dQ}{dP}(\mathbf{x}) \right)^{-\alpha} - E_Q \left[ \left( \frac{dQ}{dP}(\mathbf{x}) \right)^{-\alpha} \right] \right\}^2 \right]$$

$$= \frac{1}{\alpha^2} \cdot E_Q \left[ \left\{ \left( \frac{dQ}{dP}(\mathbf{x}) \right)^{-\alpha} \right\}^2 \right] - \frac{1}{\alpha^2} \cdot \left\{ E_Q \left[ \left( \frac{dQ}{dP}(\mathbf{x}) \right)^{-\alpha} \right] \right\}^2$$

$$= \frac{1}{\alpha^2} \cdot E_P \left[ \frac{dQ}{dP}(\mathbf{x}) \cdot \left( \frac{dQ}{dP}(\mathbf{x}) \right)^{-2\alpha} \right] - \frac{1}{\alpha^2} \cdot \left\{ E_P \left[ \frac{dQ}{dP}(\mathbf{x}) \cdot \left( \frac{dQ}{dP}(\mathbf{x}) \right)^{-\alpha} \right] \right\}^2$$

$$= \frac{1}{\alpha^2} \cdot E_P \left[ \left( \frac{dQ}{dP}(\mathbf{x}) \right)^{1-2\alpha} \right] - \frac{1}{\alpha^2} \cdot \left\{ E_P \left[ \left( \frac{dQ}{dP}(\mathbf{x}) \right)^{1-\alpha} \right] \right\}^2$$

$$= \frac{1}{\alpha^2} \cdot \left\{ 2\alpha \cdot (2\alpha - 1) \cdot \left( \frac{1}{2\alpha \cdot (2\alpha - 1)} \cdot E_P \left[ \left( \frac{dQ}{dP}(\mathbf{x}) \right)^{1-2\alpha} - 1 \right] \right) + 1 \right.$$

$$- \alpha^2 \cdot (1-\alpha)^2 \cdot \left( \frac{1}{\alpha \cdot (\alpha - 1)} \cdot E_P \left[ \left( \frac{dQ}{dP}(\mathbf{x}) \right)^{1-\alpha} - 1 \right] \right)^2 + 1$$

$$+ \alpha^2 \cdot (1-\alpha)^2 \cdot \left( \frac{2}{\alpha \cdot (\alpha - 1)} \cdot E_P \left[ \left( \frac{dQ}{dP}(\mathbf{x}) \right)^{1-\alpha} - 1 \right] \right)$$

$$\left. - 2\alpha \cdot (1-\alpha) \right\}. \tag{71}$$

Similarly, let $L_P^i = \frac{1}{1-\alpha} \cdot \left\{ \left( \frac{dQ}{dP}(\mathbf{X}_P^i) \right)^{1-\alpha} - E_P \left[ \left( \frac{dQ}{dP}(\mathbf{x}) \right)^{1-\alpha} \right] \right\}$. Then $\{L_P^i\}_{i=1}^N$ are independent and identically distributed variables whose means and variances are as follows:

$$E_P \left[ L_P^i \right] = 0, \tag{72}$$

and

$$\mathrm{Var}_P\left[L_P^i\right]$$

$$= E_P\left[\frac{1}{(1-\alpha)^2}\cdot\left\{\left(\frac{dQ}{dP}(\mathbf{x})\right)^{1-\alpha}-E_P\left[\left(\frac{dQ}{dP}(\mathbf{x})\right)^{1-\alpha}\right]\right\}^2\right]$$

$$= \frac{1}{(1-\alpha)^2}\cdot E_P\left[\left\{\left(\frac{dQ}{dP}(\mathbf{x})\right)^{1-\alpha}\right\}^2\right]-\frac{1}{(1-\alpha)^2}\cdot\left\{E_P\left[\left(\frac{dQ}{dP}(\mathbf{x})\right)^{1-\alpha}\right]\right\}^2$$

$$= \frac{1}{(1-\alpha)^2}\cdot E_P\left[\left(\frac{dQ}{dP}(\mathbf{x})\right)^{2(1-\alpha)}\right]-\frac{1}{(1-\alpha)^2}\cdot\left\{E_P\left[\left(\frac{dQ}{dP}(\mathbf{x})\right)^{1-\alpha}\right]\right\}^2$$

$$= \frac{1}{(1-\alpha)^2}\cdot E_P\left[\left(\frac{dQ}{dP}(\mathbf{x})\right)^{1-(2\alpha-1)}\right]-\frac{1}{(1-\alpha)^2}\cdot\left\{E_P\left[\left(\frac{dQ}{dP}(\mathbf{x})\right)^{1-\alpha}\right]\right\}^2$$

$$= \frac{1}{(1-\alpha)^2}\cdot\left\{2\alpha\cdot(2\alpha-1)\cdot\left(\frac{1}{2\alpha\cdot(2\alpha-1)}\cdot E_P\left[\left(\frac{dQ}{dP}(\mathbf{x})\right)^{1-(2\alpha-1)}-1\right]\right)+1\right.$$

$$-\ \alpha^2\cdot(1-\alpha)^2\cdot\left(\frac{1}{\alpha\cdot(\alpha-1)}\cdot E_P\left[\left(\frac{dQ}{dP}(\mathbf{x})\right)^{1-\alpha}-1\right]\right)^2+1$$

$$+\ \alpha^2\cdot(1-\alpha)^2\cdot\left(\frac{2}{\alpha\cdot(\alpha-1)}\cdot E_P\left[\left(\frac{dQ}{dP}(\mathbf{x})\right)^{1-\alpha}-1\right]\right)$$

$$\left.-\ 2\alpha\cdot(1-\alpha)\right\}. \tag{73}$$

Now, we consider an asymptotical distribution of the following term:

$$\sqrt{N}\cdot\left\{\hat{D}^{(N)}(Q||P\,;\,\alpha)-D(Q||P\,;\,\alpha)\right\}$$

$$= \sqrt{N}\cdot\left(\frac{1}{N}\cdot\sum_{i=1}^N\left\{L_Q^i-E_Q\left[L_Q^i\right]\right\}\right)+\sqrt{N}\cdot\left(\frac{1}{N}\cdot\sum_{i=1}^N\left\{L_P^i-E_P\left[L_P^i\right]\right\}\right). \tag{74}$$

By the central limit theorem, we observe that as $N\to\infty$,

$$\sqrt{N}\cdot\left(\frac{1}{N}\cdot\sum_{i=1}^N\left\{L_Q^i-E_Q\left[L_Q^i\right]\right\}\right)\quad\xrightarrow{d}\quad\mathcal{N}\big(0,\mathrm{Var}_Q\left[L_Q^i\right]\big), \tag{75}$$

and

$$\sqrt{N}\cdot\left(\frac{1}{N}\cdot\sum_{i=1}^N\left\{L_P^i-E_P\left[L_P^i\right]\right\}\right)\quad\xrightarrow{d}\quad\mathcal{N}\big(0,\mathrm{Var}_P\left[L_P^i\right]\big). \tag{76}$$

Therefore, from Equations (75) and (76), we obtain

$$\sqrt{N}\cdot\left\{\hat{D}^{(N)}(Q||P\,;\,\alpha)-D(Q||P\,;\,\alpha)\right\}\xrightarrow{d}\mathcal{N}\big(0,\sigma_\alpha^2\big), \tag{77}$$

and

$$\sigma_\alpha^2 = \mathrm{Var}_Q\left[L_Q^i\right]+\mathrm{Var}_P\left[L_P^i\right]$$
$$= C_\alpha^1\cdot D(Q||P\,;\,2\alpha)+C_\alpha^2\cdot D(Q||P\,;\,2\alpha-1)$$
$$+\ C_\alpha^3\cdot D(Q||P\,;\,\alpha)^2+C_\alpha^4\cdot D(Q||P\,;\,\alpha)+C_\alpha^5, \tag{78}$$

where

$$C_\alpha^1 = \frac{2\alpha \cdot (1 - 2\alpha)}{\alpha^2}, \tag{79}$$

$$C_\alpha^2 = \frac{2\alpha \cdot (1 - 2\alpha)}{(1 - \alpha)^2}, \tag{80}$$

$$C_\alpha^3 = -\frac{1}{\alpha^2} - \frac{1}{(1 - \alpha)^2}, \tag{81}$$

$$C_\alpha^4 = \frac{2}{\alpha^2} + \frac{2}{(1 - \alpha)^2},$$

$$C_\alpha^5 = \left( \frac{1}{\alpha^2} + \frac{1}{(1 - \alpha)^2} \right) \cdot \left( 2 - 2\alpha \cdot (1 - \alpha) \right). \tag{82}$$

This completes the proof. $\qquad\square$

## D  DETAILS OF THE EXPERIMENTS IN SECTION 6

In this section, we provide details on the hyperparameter settings used in the experiments described in Section 6.

### D.1  DETAILS OF THE EXPERIMENTS IN SECTION 6.1

In this section, we detail the experiments reported in Section 6.1.

#### D.1.1  DATASETS.

We generated the following 100 train datasets. $P = \mathcal{N}(\mu_p, I_5)$ and $Q = \mathcal{N}(\mu_q, \Sigma_q)$ where $\mu_p = \mu_q = (0, 0, \ldots, 0)$, and $I_5$ denotes the 5-dimensional identity matrix, and $\Sigma_q = (\sigma_{ij})_{i=1}^5$ with $\sigma_{ii} = 1$, and $\sigma_{ij} = 0.8$ for $i \neq j$. The size of each dataset was 5000.

#### D.1.2  EXPERIMENTAL PROCEDURE.

Neural networks were traind using the synthetic datasets by optimizing $\alpha$-Div for $\alpha = -3.0, -2.0, -1.0, 0.2, 0.5, 0.8, 2.0, 3.0,$ and $4.0$ while measuring the training losses for each learning step. For each value of $\alpha$, 100 trials were conducted. Finally, we reported the median, ranging between the 45th and 55th quartiles, and between the 2.5th and 97.5th quartiles of the training losses at each learning step.

#### D.1.3  NEURAL NETWORK ARCHITECTURE, OPTIMIZATION ALGORITHM, AND HYPERPARAMETERS.

A 5-layer perceptron with ReLU activation was used, with each hidden layer comprising 100 nodes. For optimization, the learning rate was set to 0.001, the batch size was 2500, and the number of epochs was 250. The models for DRE were implemented using the PyTorch library (Paszke et al., 2017) in Python. Training was conducted with the Adam optimizer (Kingma, 2014) in PyTorch and an NVIDIA T4 GPU.

#### D.1.4  RESULTS.

As shown in Figure 4, the training losses of $\alpha$-Div across learning steps are presented for $\alpha = -3, -2, -1, 0.2, 0.5, 0.8, 2.0, 3.0,$ and $4.0$. The upper ($\alpha = -3.0, -2.0,$ and $-1.0$) and middle ($\alpha = 2.0, 3.0$ and $4.0$) figures in Figure 4show that the training losses diverged to large negative values when $\alpha < 0$ or $\alpha > 1$. In contrast, the bottom figure ($\alpha = 0.2, 0.5,$ and $0.8$) Figure 1, the training losses of $\alpha$-Div converged, illustrating the stability of optimization with $\alpha$-Div when $0 < \alpha < 1$.

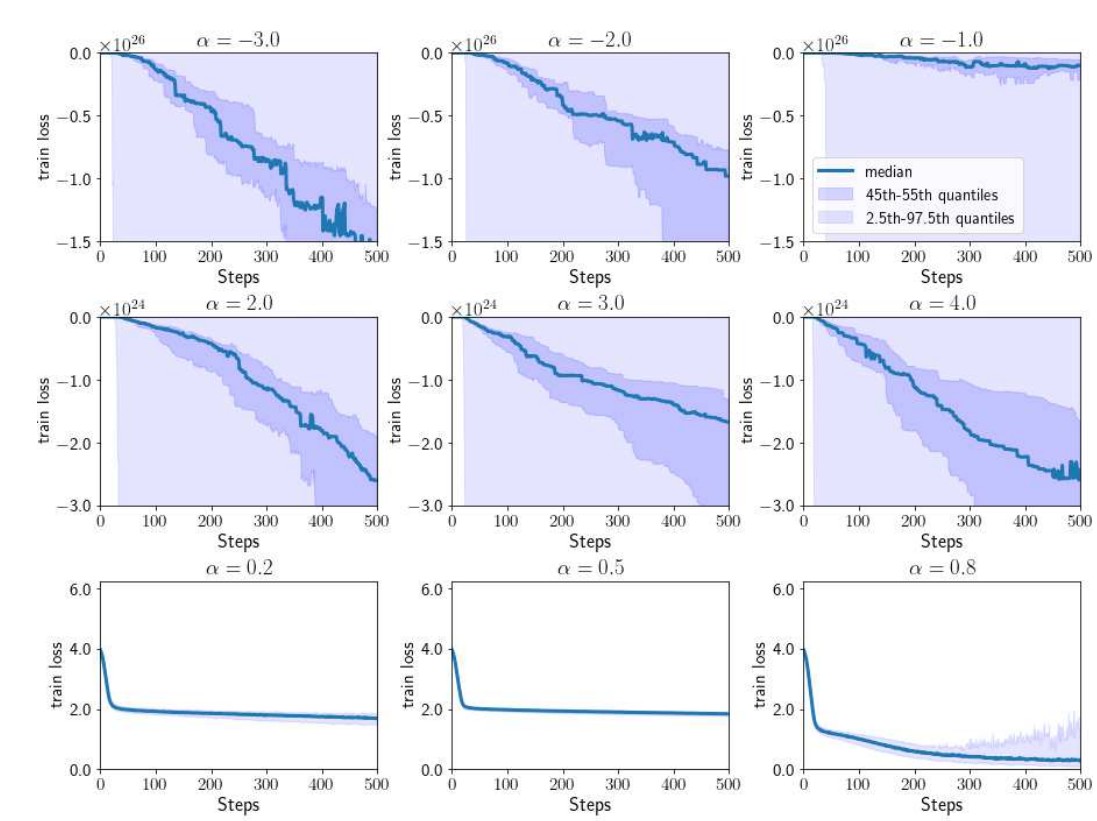

Figure 4: All results of Section 6.1. for $\alpha = -3, -2, -1, 0.2, 0.5, 0.8, 2.0, 3.0,$ and $4.0$. Each graph displays the training losses (y-axis) against the learning steps (x-axis) during optimization using $\alpha$-Div for different values of $\alpha$ values. The solid blue line represents the median training losses. The dark blue area indicates the range between the 45th and 55th percentiles, while the light blue area shows the range between the 2.5th and 97.5th percentiles of the training losses.

## D.2 DETAILS OF THE EXPERIMENTS IN SECTION 6.2

In this section, we provide details about the experiments reported in Section 6.2.

### D.2.1 DATASETS.

We first generated 100 training datasets, each with a total size of 10000 samples. Each dataset was drawn from two normal distributions: $P = \mathcal{N}(\mu_p, \cdot I_5)$ and $Q = \mathcal{N}(\mu_q, \cdot I_5)$ where $I_5$ denotes the 5-dimensional identity matrix, and $\mu_p = (-5/2, 0, 0, 0, 0, 0)$ and $\mu_q = (5/2, 0, 0, 0, 0, 0)$.

### D.2.2 EXPERIMENTAL PROCEDURE.

We trained neural networks using the training datasets, optimizing both $\alpha$-Div and the standard $\alpha$-divergence loss function defined in Equation (7) with $\alpha = 0.5$, as well as nnBD-LSIF, while measuring the training losses at each learning step. We conducted 100 trials and reported the median

training losses, along with the ranges between the 45th and 55th percentiles, and between the 2.5th and 97.5th percentiles, at each learning step.

**Loss functions used in the experiments.** We used $\alpha$-Div, the standard $\alpha$-divergence loss function, and the non-negative Bregman divergence least-squares importance fitting (nnBD-LSIF) loss function (Kato & Teshima, 2021) to train neural networks. The standard $\alpha$-divergence loss function, presented in Equation (7), exhibits a biased gradient when $\alpha < 1$.

nnBD-LSIF is an unbounded Bregman divergence loss function obtained from the deep direct DRE (D3RE) method proposed by Kato & Teshima (2021), which is defined as

$$\mathcal{L}_{\text{nnBD-LSIF}}(\phi) = -\hat{E}_Q \left[ \phi(\mathbf{x}) - \frac{C}{2} \phi^2(\mathbf{x}) \right] + \left( \frac{1}{2} \cdot \hat{E}_P \left[ \phi^2(\mathbf{x}) \right] - \frac{C}{2} \cdot \hat{E}_Q \left[ \phi^2(\mathbf{x}) \right] \right)_+, \quad (83)$$

where $(a)_+ = a$ if $a > 0$ otherwise $(a)_+ = 0$ and $C$ is positive constant. Note that, nnBD-LSIF has a unbiased gradient. The optimization efficiency of nnBD-LSIF was observed to confirm the effectiveness of an unbiased gradient of an $f$-divergence loss function as well as $\alpha$-Div.

### D.2.3 NEURAL NETWORK ARCHITECTURE, OPTIMIZATION ALGORITHM, AND HYPERPARAMETERS.

We used a 4-layer perceptron with ReLU activation, where each hidden layer contained 100 nodes. For optimization, the learning rate was set to 0.00005, the batch size was 2500, and the number of epochs was 1000. We implemented all models for DRE using the PyTorch library (Paszke et al., 2017) in Python. Training was performed with the Adam optimizer (Kingma, 2014) in PyTorch, utilizing an NVIDIA T4 GPU.

### D.3 DETAILS OF THE EXPERIMENTS IN SECTION 6.3

In this section, we provide details on the experiments reported in Section 6.3.

### D.3.1 DATASETS.

Initially, we created 100 train and test datasets, each with a size of 10,000. Each dataset is generated from two normal distributions $P = \mathcal{N}(\mu_p, \sigma^2 \cdot I_3)$ and $Q = \mathcal{N}(\mu_q, 4^2 \cdot I_3)$ where $I_3$ denotes the 3-dimensional identity matrix and $\sigma$ values were 1.0, 1.1, 1.2, 1.4, 1.6, 2.0, 2.5, or 3.0, and $\mu_p = (-3/2, -3/2, -3/2)$ and $\mu_q = (3/2, 3/2, 3/2)$. In the aforementioned setting, the ground truth KL-divergence amounts of the datasets is obtained as

$$
\begin{aligned}
KL(P\|Q) &= E_P \left[ \log \left( \frac{dP}{dQ} \right) \right] \\
&= \frac{1}{2} \cdot \left[ \log \frac{|\Sigma_p|}{|\Sigma_q|} - d + \text{Tr}(\Sigma_p^{-1} \cdot \Sigma_q) + (\mu_p - \mu_q)^T \cdot \Sigma_p^{-1} \cdot (\mu_p - \mu_q) \right] \\
&= \frac{1}{2} \cdot \left[ \log \frac{\sigma^2 \cdot |I_3|}{4^2 \cdot |I_3|} - 3 + \text{Tr}(\sigma^{-2} \cdot I_3 \cdot 4^2 \cdot I_3) + 3 \cdot \mathbf{1}^T \cdot \sigma^{-2} \cdot I_3 \cdot 3 \cdot \mathbf{1} \right] \\
&= \frac{1}{2} \cdot \left( 6 \log \sigma - 12 \log 2 - 3 + 3 \cdot \sigma^{-2} \cdot 16 + 27 \cdot \sigma^{-2} \right) \\
&= 3 \log \sigma - 6 \log 2 - \frac{3}{2} + \sigma^{-2} \cdot \frac{75}{2}. \quad (84)
\end{aligned}
$$

From Equation (84), we see that the ground truth KL-divergence amounts of the datasets were 31.8, 25.6, 21.0, 14.5, 10.4, 10.4, 5.8, 3.1, and 1.8, which correspond to the ascending $\sigma$ values, such that $\sigma = 1.0, 1.1, ...3.0$.

### D.3.2 EXPERIMENTAL PROCEDURE.

We trained neural networks using the training datasets by optimizing both $\alpha$-Div with $\alpha = 0.5$ and a KL-divergence loss function. Details of the KL-divergence loss function used in the experiments are provided in the following paragraph. Training was halted if the validation losses, measured using the

validation datasets, did not improve during an entire epoch. After training the neural networks, we measured the root mean squared error (RMSE) of the estimated density ratios using the test datasets. We estimated the KL-divergence of the test datasets for each trial using the estimated density ratios and the plug-in estimation method, which is detailed below. A total of 100 trials were conducted. Finally, we reported the median RMSE of the DRE and the estimated KL-divergence, along with the interquartile range (25th to 75th percentiles), for each KL-divergence loss function and $\alpha$-Div.

**KL-divergence loss function.**  A standard KL-divergence loss function is obtained as

$$\mathcal{L}_{\text{standard-KL}}(\phi) = \hat{E}_P[\phi] - \hat{E}_Q[\log \phi]. \tag{85}$$

In our pre-experiment, the standard KL-divergence loss function exhibited poor optimization performance, which we attribute to its biased gradients. However, we found that the Gibbs density transformation, as described in Section 4.1, improved optimization performance for the KL-divergence loss function. Therefore, we used the following KL-divergence loss function, $\mathcal{L}_{\text{KL}}(\cdot)$ in our experiments:

$$\mathcal{L}_{\text{KL}}(T) = \hat{E}_P[e^T] - \hat{E}_Q[T]. \tag{86}$$

**Plug-in KL-divergence estimation method using the estimated density ratios.**  The KL-divergence of the test datasets was estimated by estimated predicted density ratios for the test datasets using a plug-in estimation, such that

$$\widehat{KL}(P||Q) = \hat{E}_Q[\log \hat{r}_q(\mathbf{x})], \tag{87}$$

where $\hat{r}_q(\mathbf{x}) = e^{T(\mathbf{x})}/\hat{E}_Q[e^{T(\mathbf{x})}]$.

### D.3.3 Neural Network Architecture, Optimization Algorithm, and Hyperparameters.

The same neural network architecture, optimization algorithm, and hyperparameters were used for both $\alpha$-Div and the KL-divergence loss function. A 4-layer perceptron with ReLU activation was employed, with each hidden layer consisting of 256 nodes. For optimization with the $\alpha$-Div loss function, the value of $\alpha$ was set to 0.5, the learning rate was 0.00005, and the batch size was 256. Early stopping was applied with a patience of 32 epochs, and the maximum number of epochs was set to 5000. For optimization using the KL-divergence loss function, the learning rate was 0.00001, with a batch size of 256. Early stopping was applied with a patience of 2 epochs, and the maximum number of epochs was 5000. Pytorch (Paszke et al., 2017) library in Python was used to implement all models for DRE, with the Adam optimizer (Kingma, 2014) in PyTorch and an NVIDIA T4 GPU used for training the neural networks.

## E Additional Experiments

### E.1 Comparison with Existing DRE Methods

We empirically compare the proposed DRE method with existing DRE methods in terms of accuracy in DRE tasks. This experiment followed the setup described in Kato & Teshima (2021).

### E.1.1 Existing $f$-Divergence Loss Functions for Comparison.

The proposed method was compared with the Kullback-Leibler importance estimation procedure (KLIEP) (Sugiyama et al., 2007b), unconstrained least-squares importance fitting (uLSIF) (Kanamori et al., 2009), and deep direct DRE (D3RE) (Kato & Teshima, 2021). The `densratio` library in R was used for KLIEP and uLSIF.[2] For D3RE, the non-negative Bregman divergence least-squares importance fitting (nnBD-LSIF) loss function was employed.

### E.1.2 Datasets.

For each $d = 10, 20, 30, 50$, and $100$, 100 datasets were generated, comprising training and test sets drawn from two $d$-dimensional normal distributions $P = \mathcal{N}(\mu_p, I_d)$ and $Q = \mathcal{N}(\mu_q, I_d)$, where $I_d$ denotes the $d$-dimensional identity matrix, $\mu_p = (0, 0, \ldots, 0)$, and $\mu_q = (1, 0, \ldots, 0)$.

---

[2]https://cran.r-project.org/web/packages/densratio/index.html

Table 5: Results of additional experiments described in Section E.1. The table reports the average mean and standard deviation of the MSE for DRE with each method. Results are presented in the format "mean (standard deviation)." The lowest MSE values are highlighted in bold.

| | Data dimentions ($d$) | | | | |
|---|---|---|---|---|---|
| Model | $d = 10$ | $d = 20$ | $d = 30$ | $d = 50$ | $d = 100$ |
| KLIEP | 2.141(0.392) | 2.072(0.660) | 2.005(0.569) | 1.887(0.450) | 1.797(0.419) |
| uLSIF | 1.482(0.381) | 1.590(0.562) | 1.655(0.578) | 1.715(0.446) | 1.668(0.420) |
| D3RE | 1.111(0.314) | 1.127(0.413) | 1.219(0.458) | 1.222(0.305) | 1.369(0.355) |
| $\alpha$-Div | **0.173(0.072)** | **0.278(0.113)** | **0.479(0.259)** | **0.665(0.194)** | **1.118(0.314)** |

### E.1.3 EXPERIMENTAL PROCEDURE.

Model parameters were trained using the training datasets, and density ratios for the test datasets were estimated. The mean squared error (MSE) of the estimated density ratios for the test datasets was calculated based on the true density ratios. Finally, the average mean and standard deviation of the MSE for each method were reported.

### E.1.4 NEURAL NETWORK ARCHITECTURE, OPTIMIZATION ALGORITHM, AND HYPERPARAMETERS.

For both D3RE and $\alpha$-Div, a 3-layer perceptron with 100 hidden units per layer was used, consistent with the neural network structure employed in Kato & Teshima (2021). For D3RE, the learning rate was set to 0.00005, the batch size was 128, and the number of epochs was 250 for each data dimension. The hyperparameter $C$ was set to 2.0. For $\alpha$-Div, the learning rate was set to 0.0001, the batch size was 128, and the value of $\alpha$ was set to 0.5 for each data dimension. The number of epochs was set to 40 for data dimensions of 10, 50 for dimensions of 20, 30, and 50, and 60 for a dimension of 100. The PyTorch library (Paszke et al., 2017) in Python was used to implement all models for both D3RE and $\alpha$-Div. The Adam optimizer (Kingma, 2014) in PyTorch, along with an NVIDIA T4 GPU, was used for training the neural networks.

**Results.** Table 5 summarizes the results for each method across different data dimensions. Six cases where the MSE for KLIEP exceeded 1000 were excluded. For all data dimensions, $\alpha$-Div consistently demonstrated superior accuracy compared to the other methods, achieving the lowest MSE values. However, it is important to note that the prediction accuracy of $\alpha$-Div significantly decreased as the data dimensions increased. The curse of dimensionality in DRE was also observed in experiments with real-world data, which will be reported in the next section.

## E.2 EXPERIMENTS USING REAL-WORLD DATA

We present numerical experiments using real-world data to highlight important considerations for applying the proposed method. Specifically, we conducted experiments on Domain Adaptation (DA) for classification models using the Importance Weighting (IW) method (Shimodaira, 2000). The IW method builds a prediction model for a target domain using data from a source domain, while adjusting the distribution of source domain features to match that of the target domain features by employing the density ratio between the source and target domains as sample weights.

In these experiments, we used Amazon review data (Blitzer et al., 2007) and employed two prediction algorithms: linear regression and gradient boosting. The hyperparameters for each algorithm were selected from a predefined set based on validation accuracy, using the Importance Weighted Cross Validation (IWCV) method (Sugiyama et al., 2007a) on the source domain data.

Through these experiments, we observed a decline in prediction accuracy on test data from the target domain as the data dimensionality increased. Specifically, there were instances where the accuracy worsened compared to models that did not use importance weighting—i.e., models trained solely on the source data. These phenomena are likely due to two issues in DRE: the degradation in density ratio estimation accuracy as dimensionality increases, as noted in Section E.1, and the negative

impact of high KL-divergence on density ratio estimation, as observed in Section 6.3. It is important to note that KL-divergence increases with the number of features (i.e., data dimensions), unless all features are fully independent.

### E.2.1 DATASETS.

The Amazon review dataset (Blitzer et al., 2007) includes text reviews and rating scores from four domains: books, DVDs, electronics, and kitchen appliances. The text reviews are one-hot encoded, and the rating scores are converted into binary labels. Twelve domain adaptation classification tasks were conducted, with each domain serving once as the source domain and once as the target domain.

**Notation.** $\mathbf{X}_S^d$ and $\mathbf{X}_T^d$ denote subsets of the original data for the source and target domains, respectively, for each feature dimension $d$, where the columns of $\mathbf{X}_S^d$ and $\mathbf{X}_T^d$ are identical. $y_S$ and $y_T$ represent the objective variables in the source and target domains, respectively, which are binary labels assigned to each sample in the source and target domain data. $\mathbf{Z}_S^d$ and $\mathbf{Z}_T^d$ denote $d$-dimensional feature tables used to estimate the density ratio $\hat{r}(\mathbf{Z}_S^d)$, which is the ratio of the target domain density to the source domain density. $\dim(X)$ indicates the number of columns (features) in the data $X$.

### E.2.2 EXPERIMENTAL PROCEDURE.

**Step 1. Creation of feature tables.** Many DA methods utilize feature embedding techniques to project high-dimensional data into a lower-dimensional feature space, facilitating the handling of distribution shifts between source and target domains (Ragab et al., 2023). However, our preliminary experiments revealed that model prediction accuracies were significantly influenced by the embedding procedures. To address these effects on DA task accuracies, we explored an embedding method with theoretical considerations detailed in the next section.

Specifically, we selected an identical set of columns from the original data of both the source and target domains for each feature dimension, $d = 8, 16, 32, 64$, and $128$, arranging the columns in ascending order of $d$. Let $\mathbf{X}_S^d$ and $\mathbf{X}_T^d$ denote the subsets of the original data for the source and target domains, respectively, determined by these selected columns for each $d$. We then generated a $d \times \dim(\mathbf{X}_S^d)$ matrix $A_d$ from a normal distribution. Finally, by multiplying $\mathbf{X}_S^d$ and $A_d$, and $\mathbf{X}_T^d$ and $A_d$, we obtained the feature tables $\mathbf{Z}_S^d$ and $\mathbf{Z}_T^d$, embedding the original source and target domain data into a $d$-dimensional feature space. [3]

**Step 2. Estimation of importance weights.** Using the proposed loss function with $\mathbf{Z}_S^d$ and $\mathbf{Z}_T^d$ obtained from the previous step, we estimated the probability density ratio $\hat{r}(\mathbf{Z}_S^d)$ for each feature dimension $d$, where $r(\mathbf{Z}_S^d) = q(\mathbf{Z}_T^d)/p(\mathbf{Z}_S^d)$. This ratio represents the density of the target domain relative to the source domain.

**Step 3. Model construction.** We constructed the target model using the IW method. Specifically, we built a classification model using the training dataset $(\mathbf{X}_S^d, y_S)$, where the estimated density ratio $\hat{r}(\mathbf{Z}_S^d)$ served as the sample weights for the IW method. Additionally, we constructed a prediction model using only the source data, i.e., a model built without importance weighting.

**Step 4. Verification of prediction accuracy** To evaluate the prediction accuracy of the models, we selected the ROC AUC score, as it measures accuracy independently of the thresholds used for label determination. For the classification tasks in domain adaptation, we employed two classification methods, each representing a different algorithmic approach: `LogisticRegression` from the `scikit-learn` library (Pedregosa et al., 2011) for linear classification, and `LightGBM` (Ke et al., 2017) for nonlinear classification.

The hyperparameter sets for both methods used to evaluate prediction accuracy on target domains were selected using the IWCV method (Sugiyama et al., 2007a). These hyperparameter sets were

---

[3]In our experiments, we utilized matrices generated from the normal distribution as embedding maps, which is equivalent to random projection (Bingham & Mannila, 2001). However, the linearity of the map is not necessary for preserving the density ratios, as discussed in Section E.2.3. In contrast, linearity is a key requirement for the distance-preserving property of random projection.

defined as all combinations of the values listed in Table 6 for `LogisticRegression` and Table 7 for `LightGBM`, respectively. Finally, the prediction accuracies on the target domain were assessed using the best model selected through IWCV, with the target domain data $(\mathbf{X}_\mathrm{T}^d, y_\mathrm{T})$ used for reporting.

Table 6: Hyperparameter values for LogisticRegression. "Hyperparameters" shows the hyperparameter names used in the library. Texts inside parentheses provide explanations of the parameters.

| Hyperparameters | Values |
|---|---|
| l1_ratio (Elastic-Net mixing parameter) | 0, 0.1, 0.2, 0.3, 0.4, 0.5, 0.6, 0.7, 0.8, 0.9, and 1.0 |
| lambda (Inverse of regularization strength) | 0.0001, 0.001, 0.01, 0.05, 0.1, 0.25, 0.5, 0.75, 1, 1.5, 2, and 5 |

Table 7: Hyperparameter values for LightGBM. "Hyperparameters" shows the hyperparameter names used in the library. Texts inside parentheses provide explanations of the parameters.

| Hyperparameters | Values |
|---|---|
| lambda_l1 ($L_1$ regularization) | 0.0, 0.25, 0.5, 0.75, and 1.0 |
| lambda_l2 ($L_2$ regularization) | 0.0, 0.0001, 0.001, 0.01, 0.1, 0.5, 1.0, 2.0, and 4.0 |
| num_leaves (Number of leaves in trees) | 64, 248, 1024, 2048, and 4096 |
| learning_rate (Learning rate) | 0.01, and 0.001 |
| feature_fraction (Ratio of featuresr used for modeling) | 0.4, 0.8, and 1.0 |

Table 8: Original data dimensions ($\dim(\mathbf{X})$) used to obtain feature dimensions ($\dim(\mathbf{Z})$) by embedding.

| | Feature dimensions ($d = \dim(\mathbf{Z})$) | | | | |
|---|---|---|---|---|---|
| | $d = 8$ | $d = 16$ | $d = 32$ | $d = 64$ | $d = 128$ |
| Original data dimensions ($\dim(\mathbf{X})$) | 500 | 700 | 900 | 1700 | 4600 |

### E.2.3 CONSIDERATION OF THE FEATURE EMBEDDING METHOD

Let $f : \mathbf{X} \longmapsto \mathbf{Z}$ denote a $C^1$-class embedding map which transforms the original data $\mathbf{X} \subseteq \mathbb{R}^{N \times D}$ to the feature table $\mathbf{Z} \subseteq \mathbb{R}^{N \times d}$ with $d < D$.

We now demonstrate that if $f$ is injective for both the source and target domain data, it preserves the density ratio between the target and source domain densities when mapping from the original data to the embedded data.

To see this, we use the singular value decomposition (SVD) of the Jacobian matrix $J_f(\mathbf{x})$ of $f$, which gives

$$J_f(\mathbf{x}) = U(\mathbf{x}) \cdot \Sigma(\mathbf{x}) \cdot V^T(\mathbf{x})$$

with

$$\Sigma(\mathbf{x}) = \begin{pmatrix} \sigma_1(\mathbf{x}) & 0 & \ldots & 0 \\ 0 & \sigma_2(\mathbf{x}) & \ldots & 0 \\ \vdots & \vdots & \ddots & \vdots \\ 0 & 0 & \ldots & \sigma_{\dim(\mathbf{z})}(\mathbf{x}) \\ 0 & 0 & \ldots & 0 \\ \vdots & \vdots & \ldots & \vdots \\ 0 & 0 & \ldots & 0, \end{pmatrix}$$

where $U(\mathbf{x})$ and $V^T(\mathbf{x})$ are orthogonal matrices in $\mathbb{R}^{\dim(\mathbf{x})\times\dim(\mathbf{x})}$ and $\mathbb{R}^{\dim(\mathbf{z})\times\dim(\mathbf{z})}$, respectively, and $\sigma_i(\mathbf{x}) \neq 0$ for all $i$. This leads to the following relationship for the probability densities between the original and embedded data:

$$p_{\mathbf{X}}(\mathbf{x}) = \left( \prod_{i=1}^{\dim(\mathbf{z})} \sigma_i(\mathbf{x}) \right) p_{\mathbf{Z}}(f(\mathbf{x})). \tag{88}$$

From Equation (88), the probability density ratio between the source and target domains of data embedded by $f$ is obtained as

$$\frac{q_{\mathbf{X}}(\mathbf{x})}{p_{\mathbf{X}}(\mathbf{x})} = \frac{\left( \prod_{i=1}^{\dim(\mathbf{z})} \sigma_i(\mathbf{x}) \right) \cdot q_{\mathbf{Z}}(f(\mathbf{x}))}{\left( \prod_{i=1}^{\dim(\mathbf{z})} \sigma_i(\mathbf{x}) \right) \cdot p_{\mathbf{Z}}(f(\mathbf{x}))} = \frac{q_{\mathbf{Z}}(\mathbf{z})}{p_{\mathbf{Z}}(\mathbf{z})}.$$

Therefore, $f$ preserves the density ratio from the original data to the embedded data. Additionally, if $f$ is a matrix multiplication, its injectivity can be achieved for $\mathbf{X}$ by reducing its dimensionality sufficiently. Reducing the dimensionality of $\mathbf{X}$ can induce the injectivity of $f$.

We heuristically detected the injectivity of our embedding by observing the following: We identified the largest subset of columns in $\mathbf{Z}_S^d$ such that a sufficient increase in the KL divergence between $P(\mathbf{Z}_S^d)$ and $P(\mathbf{Z}_T^d)$ increased significantly as the number of columns increased. Injectivity was assumed for columns within this subset.

Although our feature embedding procedure is based on heuristic observations and lacks thorough theoretical analysis, we find it adequate for evaluating the performance of the proposed method in DRE downstream tasks with real-world data as the number of features increases.

The number of columns in the original data used in the experiments is listed in Table 8.

**Neural Network Architecture, Optimization Algorithm, and Hyperparameters.** A 5-layer perceptron with ReLU activation was used, with each hidden layer consisting of 256 nodes. For optimization, the value of $\alpha$ was set to 0.5, the learning rate was 0.0001, and the batch size was 128. Early stopping was applied with a patience of 1 epoch, and the maximum number of epochs was set to 5000. The PyTorch library (Paszke et al., 2017) in Python was used to implement all models for DRE. The Adam optimizer (Kingma, 2014) in PyTorch, along with an NVIDIA T4 GPU, was used to train the neural networks.

**Results.** The results are shown in Figure 5 (LogisticRegression) and Figure 6 (LightGBM). The domain names at the origin of the arrows in the figure titles represent the source domains, while those at the tip represent the target domains. The x-axis of each figure shows the number of features, and the y-axis represents the ROC AUC for the domain adaptation tasks. The orange line (SO) represents models trained using source-only data, i.e., models trained using source data without importance weighting, while the blue line (IW) represents models trained using source data with importance weighting.

Prediction accuracy for the models trained solely on the source data improved as the number of features increased, which is expected since more features typically lead to better accuracy. However, for both Logistic Regression and LightGBM, the performance of the IW method deteriorated as the

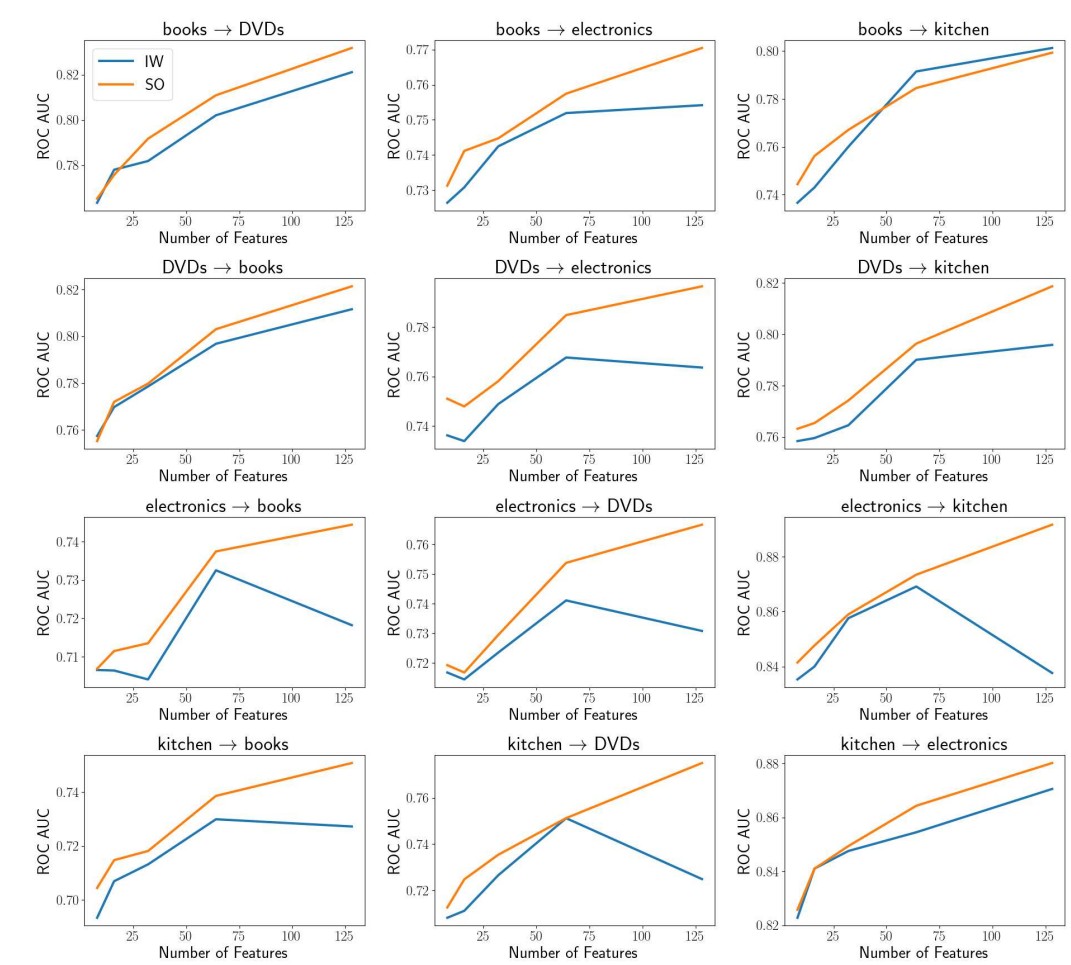

Figure 5: Results of Section E.2 for LogisticRegression. In the figure titles, domain names at the origin of the arrows indicate the source domains, while those at the tip represent the target domains. The x-axis shows the number of features, and the y-axis represents the ROC AUC for the domain adaptation tasks. The orange line (SO) denotes models trained using source-only data (i.e., models trained on source data without importance weighting), while the blue line (IW) represents models trained using source data with importance weighting.

number of features increased. A more significant decline in performance with increasing features was observed for most domain adaptation (DA) tasks, except for "books → DVDs" and "kitchen → DVDs". These results suggest that the distributions shifted by the estimated density ratios using the proposed method diverged further from the target domain as the number of features increased. Consequently, the accuracy of the density ratio estimation (DRE) likely worsened with more features.

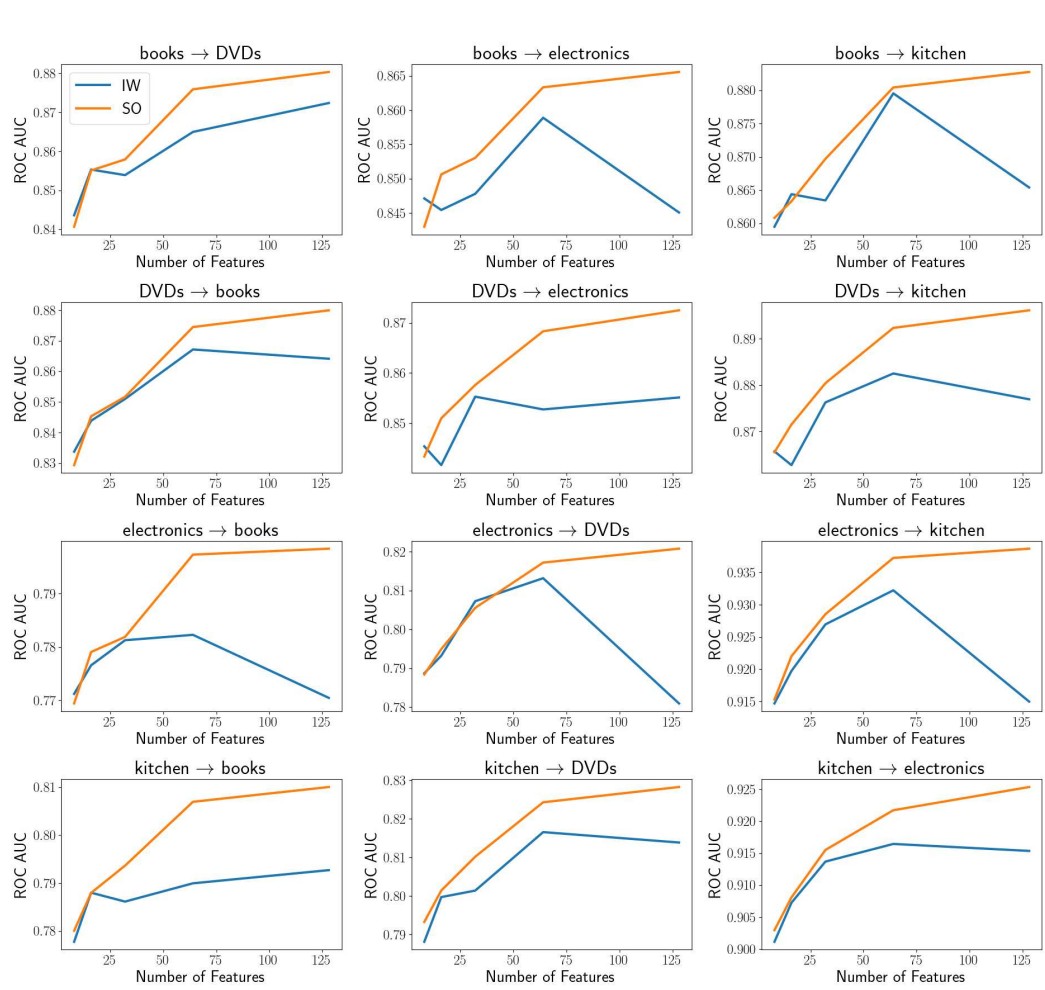

Figure 6: Results of Section E.2. Results of Section E.2 for LightGBM. In the figure titles, domain names at the origin of the arrows represent the source domains, while those at the tip indicate the target domains. The x-axis shows the number of features, and the y-axis represents the ROC AUC for the domain adaptation tasks. The orange line (SO) denotes models trained using source-only data (i.e., models trained on source data without importance weighting), whereas the blue line (IW) represents models trained using source data with importance weighting.

