# OpenReview forum: "$\alpha$-Divergence Loss Function for Neural Density Ratio Estimation"
_ICLR.cc/2025/Conference — Submitted to ICLR 2025_

### Official Review · Reviewer_qZd3 · 2024-10-30

**Soundness:** 2
**Presentation:** 3
**Contribution:** 3
**Rating:** 3
**Confidence:** 4

**Summary:**

The paper presents a new $\alpha$-divergence-based method for density ratio estimation (DRE). Compared to existing DRE methods, the presented method addresses optimization challenges such as overfitting due to lower-unbounded loss functions, vanishing training loss gradients, etc. Numerical simulations demonstrate the effectiveness of the presented method.

**Strengths:**

The paper is easy to follow.

Stable and accurate DRE is essential for machine learning, especially for GANs. The authors have provided some valuable and original research results.

Conventional DRE methods, as well as the presented $\alpha$-Div, are compared through the lens of lower boundedness and the vanishing gradient problem.

**Weaknesses:**

The paper should be revised carefully. There are many typing errors. For example, see $\hat E P[R]$ and $\hat E P[\phi]$ in Line 107 and the word "simbole" in the title of Table 2. Also, either $E_{Q}[\phi^{\alpha}]$ in Eq. (6) or $E_{Q}[e^{\alpha T}]$ in Eq. (8) is incorrect.

Some important statements are questionable. See Questions below.

Experiments are only done on simulations. It would be great if the presented method could be varied in practical scenarios associated with GANs.

**Questions:**

Line 161 says that $\nabla_{\theta} E_{Q}[log \phi_{\theta}] = E_{Q}[\nabla_{\theta} log \phi_{\theta}]$ does not hold because $E_{Q}$ is an integral. Why? Note that the distribution $Q$ is independent of $log \phi_{\theta}$. Similarly, other statements about the biased gradient problem are questionable.

Line 193 states that "These results demonstrate that major f-divergence loss functions satisfy the conditions for Equation (3)", which is clearly not true. See the KL and Peason $\chi^2$ cases.

Where is the proof of Eq. (4)?

---

> ### Author Response · Authors · 2024-12-03
>
> Due to personal circumstances, we required additional time to prepare this response, and we sincerely apologize for the delay in addressing your inquiry.
>
> We fully understand that this delay may leave you with limited time to review our response. Nevertheless, as a token of our gratitude for the time and effort you have invested in reviewing our work, we have provided detailed and thoughtful answers to the questions you raised.
>
> ---
>
> > [Q1] The paper should be revised carefully. There are many typing errors. For example, see  and  in Line 107 and the word "simbole" in the title of Table 2.
>
> [A1]
>
> We sincerely apologize for the numerous typographical errors. We have made corrections in the revised version.
>
> ---
>
> > [Q2] Also, either in Eq. (6) or  in Eq. (8) is incorrect.
>
> [A2]
>
> Thank you for your observation. We believe that both equations are correct.
>
> Equation (8) is derived by reformulating Equation (6) into the form of Gibbs density, as explained in lines 264-280 of the previous manuscript (updated to lines 254-257 of the revised version).
>
> ---
>
> >> Some important statements are questionable. See Questions below.
>
> > [Q3-1]Line 161 says that  does not hold because  is an integral. Why? Note that the distribution  is independent of . Similarly, other statements about the biased gradient problem are questionable.
>
> [A3-1]
>
> We sincerely thank the reviewer for their insightful comment and apologize for the insufficient explanation provided in the original manuscript. In response, we have added further clarification in lines 167-174 of the revised manuscript.
>
> To address the reviewer's question, below we reiterate the explanation provided in response to the review Htnn's question [Q6] and  the review 2nWh's question [Q2-2], as they are relevant to this matter.
>
>
> First, we consider a case where the interchangeability of $E$ and the differential operator $\nabla$ generally does not hold for the KL-divergence loss.
>
> Let $\phi\_{\theta} =  |x - \theta| $, where $x \in [0,1]$ and $\theta \in (0,1)$.
> The loss function is defined as: $\mathcal{L}\_{KL}(\phi\_{\theta}) = - \hat{E}_Q\big[\log \phi\_{\theta}\big] + \hat{E}_P\big[\phi\_{\theta}\big] - 1$, and the gradient is expressed as $\nabla\_{\theta} \mathcal{L}\_{KL}(\phi\_{\theta}) = -\hat{E}_Q\big[ \nabla\_{\theta}(\log \phi\_{\theta})\big] + \hat{E}_P\big[\nabla\_{\theta}(\phi\_{\theta})\big]$.
>
> The expectation of the loss is:
>
> $$
>  E\big[ \mathcal{L}\_{KL}(\phi\_{\theta})\big] = - E_Q\big[\log \phi\_{\theta}\big] + E_P\big[\phi\_{\theta}\big] - 1,
> $$
>
> and the expectation of the gradient is:
>
> $$
>  E\big[ \nabla\_{\theta} \mathcal{L}\_{KL}(\phi\_{\theta})\big] = - E_Q\big[ \nabla\_{\theta}(\log \phi\_{\theta})\big] + E_P\big[\nabla\_{\theta}(\phi\_{\theta})\big].
> $$
>
> Now, we compute
> $$
> \frac{\partial}{\partial \theta } E\big[\log \phi\_{\theta}(x)\big] = \frac{\partial}{\partial \theta } \int\_{0}^{1} \log |x - \theta| \, dx = - \log(1 - \theta).
> $$
> while:
>
> $$E\bigg[\frac{\partial}{\partial \theta } \log  \phi\_{\theta}(x)   \bigg] =  \int\_{0}^{\theta} \frac{1}{\theta-x} dx   + \int\_{\theta}^{1} \frac{1}{x-\theta} dx = \infty.
> $$
>
> Thus, $\nabla\_{\theta} E_Q\big[\log \phi\_{\theta}\big] \neq E_Q\big[\nabla\_{\theta}(\log \phi\_{\theta})\big]$.
>
> This discrepancy demonstrates that
>
> $$
> \nabla\_{\theta}  E\big[ \mathcal{L}\_{KL}(\phi\_{\theta})\big] \neq E\big[ \nabla\_{\theta} \mathcal{L}\_{KL}(\phi\_{\theta})\big].
> $$
>
> Next, we discuss this issue from a theoretical perspective. The interchangeability of $E$ and the differential operator $\nabla$ generally requires the condition of the uniform integrability.
> However, the uniform integrability of the KL-divergence loss, particularly terms expressed as "$\log |\cdot|$", does not hold when the integration domain includes intervals around zero.
>
> Therefore,  it is challenging to establish mild and natural conditions under which the expectation operator $E$ and the (partial or total) derivative of the KL-divergence loss are interchangeable.
>
> We hope this explanation clarifies the rationale behind the statements in the manuscript.

---

> ### Author Response · Authors · 2024-12-03
>
> > [Q3-2] Line 193 states that "These results demonstrate that major f-divergence loss functions satisfy the conditions for Equation (3)", which is clearly not true. See the KL and Peason  cases
>
> [A3-2]
>
> We sincerely thank the reviewer for pointing out this concern and apologize for any confusion caused by the wording in our manuscript. We recognize that the explanation in this section may not have been sufficiently clear, and we appreciate the opportunity to clarify.
>
> The statement in line 193 of the previous manuscript (line 163 of the revised manuscript):
>
> *"These results demonstrate that major f-divergence loss functions satisfy the conditions for Equation (3)"*
>
> can be more precisely described as:
>
> *"These results demonstrate that major f-divergence loss functions satisfy the conditions for Equation (3) when the estimated density ratio becomes either very small or very large."*
>
> This interpretation is supported by the second column from the right in Table 2. Specifically, for the cases of KL-divergence and Pearson divergence:
>
> - **KL-divergence**:
>   When the estimated density ratio becomes very large (i.e., the "$\phi\_{\theta} \to \infty$" column), we observe that
>   $E[\nabla\_{\theta} L_f (\phi\_{\theta})] \to E_P[\nabla\_{\theta} \phi\_{\theta}]$.
>
>   *(Note: We apologize for the error in the original description. The correct notation here is $E_P$, not $E_Q$.)*
>
> - **Pearson divergence**:
>   When the estimated density ratio becomes very small (i.e., the "$\phi\_{\theta} \to 0$" column), we observe that $E[\nabla\_{\theta} L_f (\phi\_{\theta})] \to -2 \cdot E_Q[\nabla\_{\theta} \phi\_{\theta}]$.
>
> These observations confirm that the behavior of the gradients aligns with the conditions specified in Equation (3) for both the KL-divergence and Pearson divergence cases.
>
> We hope this clarification resolves any misunderstandings.
>
> ---
>
> > [Q3-3] Where is the proof of Eq. (4)?
>
> [A3-3]
>
> Thank you for your question.
>
> The derivation of Eq. (4) is not a result of our study; hence, we did not include its proof in the paper. However, for detailed information and the proof, we kindly refer you to the cited reference [B2]:
>
> [B2] Song, J., & Ermon, S. (2019). Understanding the limitations of variational mutual information estimators. arXiv preprint arXiv:1910.06222.
>
> We hope this addresses your concern.
>
> ---
>
> > [Q4] Experiments are only done on simulations. It would be great if the presented method could be varied in practical scenarios associated with GANs.
>
> [A4]
>
> We sincerely appreciate the reviewer's insightful feedback and constructive suggestions.
>
> In fact, we attempted to apply GANs and conducted experiments using the proposed loss function. However, we did not obtain favorable results when employing the commonly used training settings, such as learning rates and other hyperparameters.
>
> As confirmed by the numerical experiments presented in Section 6.2, we suspect that this outcome arises from the high efficiency of training facilitated by the unbiased nature of the proposed loss function. Achieving a balance between the Discriminator and Generator when using this loss may require knowledge distinct from that traditionally applied to biased GANs.
>
> Exploring such training methods in GANs might be regarded as one of the themes for future research, and we decided not to include it in the numerical experiments of this study.

---

### Official Review · Reviewer_Htnn · 2024-10-31

**Soundness:** 2
**Presentation:** 2
**Contribution:** 2
**Rating:** 3
**Confidence:** 4

**Summary:**

The authors identify four major issues with existing f-divergence loss functions for DRE. They discuss train-loss hacking, biased gradients, vanishing gradients and sample rates. All of these issues are theoretically analyzed. The authors introduce an $\alpha$-divergence loss function to tackle these issues and show theoretical results for it. Experiments were conducted to investigate the effectiveness of new loss function.

**Strengths:**

- The authors address important issues in DRE.
- Improving DRE is an important problem that has significant impact on other ML disciplines, e.g., OOD detection and two-sample testing.
- Interesting ideas of how to investigate possible improvements of f-divergences for DRE.

**Weaknesses:**

+ Related work for DRE should in general be discussed more elaborate and in more detail. Novelty in comparison to these works is unclear.
+ Insufficient comparison to other neural net or kernel-based DRE methods such as [1,2,3], insufficient comparison with other f-divergences mentioned in the submission.
+ Claimed issues of f-divergences loss functions do not seem to be a problem; as performance of $\alpha$-Divergence is worse than KL-divergence for DRE in the third experiment.

[1] Kato, Masahiro, and Takeshi Teshima. "Non-negative bregman divergence minimization for deep direct density ratio estimation." International Conference on Machine Learning. PMLR, 2021.
[2] Rhodes, Benjamin, Kai Xu, and Michael U. Gutmann. "Telescoping density-ratio estimation." Advances in neural information processing systems 33 (2020): 4905-4916.
[3] Menon, Aditya, and Cheng Soon Ong. "Linking losses for density ratio and class-probability estimation." International Conference on Machine Learning. PMLR, 2016.
[4] Gruber, Lukas, Holzleitner, Markus, Lehner, Johannes, Hochreiter, Sepp, and Zellinger, Werner (2024). Overcoming Saturation in Density Ratio Estimation by Iterated Regularization. arXiv preprint arXiv:2402.13891.

**Questions:**

Experiments:
  + Why are there different datasets for each experiment, how were the parameters of the distributions chosen?
  + How was $\alpha$ chosen in your experiments and how is this hyperparameter selected in general?
  + Did KL-divergence face any of the claimed issues in the third experiment?
  + Did nnBD-LSIF face any of the claimed issues in the second experiment?

Theory:
+ Biased gradients
   * I did not immediately see why the gradient for KL is biased, can you show this formally? Biased gradients are only claimed for KL, what about other f-divergences?
   * Correct me if I'm wrong, but most kernel-based methods for DRE don't suffer from biased gradients.
+ Vanishing gradients
   * I do not really understand (3): Isn't (i) already implying vanishing gradients?
   * For me it is not clear how the limits in Table 2 are computed, a more formal derivation is needed.
   * Why should (3) happen during training? Why do very small/large density ratios imply (3), in particular how does this determine the gradient in parameter space?
   * Please explain the correctness of the two equivalences in line 347 in more detail.
   * Please show the computation of the limits in Table 4 in more detail.
+ Sample rate
   * A bad sample rate is only shown for KL-divergence, what about the other f-divergences? How does this relate to fast sample rates for DRE as discussed in [4]?
   * Describe the implications of Theorem 6.2 for the sample rate in more detail.

---

> ### Author Response · Authors · 2024-12-02
>
> Due to personal circumstances, we needed some time to prepare this response.
> we sincerely apologize for the delay in addressing your inquiry.
>
> We fully understand that the delay may leave insufficient time for you to review my response.
> However, as a token of gratitude for the time and effort you have taken to review this, we would like to provide answers to the questions you have raised.
>
> ---
>
> > Weaknesses:
>
> > [Q1]: Related work for DRE should in general be discussed more elaborate and in more detail. Novelty in comparison to these works is unclear.
> > Insufficient comparison to other neural net or kernel-based DRE methods such as [1,2,3], insufficient comparison with other f-divergences mentioned in the submission.
>
> [A1]:
>
> We have addressed the reviewer's concerns by expanding Section B, "RELATED WORK," in the appendix of the revised manuscript. This section now includes additional descriptions of prior work on density ratio estimation (DRE) methods that utilize $f$-divergence. These additions aim to provide a more detailed discussion of the related works and clarify the novelty of our approach in comparison to existing methods.
>
> ---
>
> > [Q2]: Claimed issues of f-divergences loss functions do not seem to be a problem; as performance of $f$-Divergence is worse than KL-divergence for DRE in the third experiment.
>
> [A2]:
>
> We observed in this study that the issues associated with $f$-divergence loss functions are a common challenge for all $f$-divergences, including KL-divergence losses.
>
> In the experimental results presented in Section 6.3, "Experiments on the Estimation Accuracy Using High KL-Divergence Data", we suggest that the focus should be on the similarities in estimation accuracy between KL-divergence and alpha-divergence losses, rather than solely on their differences.
>
> While it is true that estimation using KL-divergence loss was more accurate than with the proposed loss, we also observed a significant deterioration in estimation accuracy for KL-divergence loss as the KL divergence increased. This trend was similarly evident with alpha-divergence loss.
>
> Therefore, we conclude that the challenges associated with $f$-divergence loss functions are not specific to the KL-divergence loss but are intrinsic to all $f$-divergences, including the proposed loss.

---

> ### Author Response · Authors · 2024-12-02
>
> > Experiments:
>
> > [Q3] Why are there different datasets for each experiment, how were the parameters of the distributions chosen?
>
> [A3]
>
> The settings for each numerical experiment were independently designed to match the purpose of each experiment, whcih were not intended to reference one another.
> Specifically,
> * in the experiments described in Section 6.1, we used data with a small number of dimensions, the same mean values, and varying covariances.
> * In the experiments described in Section 6.2, we selected data with relatively small KL divergence values with the same covariances.
> * In the experiments described in Section 6.3, we chose data that allowed for differences in KL divergence values in a small number of dimensions.
>
> The settings used in Sections 6.1 and 6.2 could have been identical.
> However, as the comparison of results across experiments is not essential for drawing valid conclusions, we believe that using different datasets does not compromise the validity of our findings.
>
> > [Q4] Did KL-divergence face any of the claimed issues in the third experiment?
>
> [A4]
>
> Yes, the KL divergence loss exhibited a significant deterioration in accuracy as the KL divergence of the data increases.
> Thus, we believe the KL-divergence loss faced the issue discussed in Section 3.5 "SAMPLE SIZE REQUIREMENT PROBLEM FOR KL-DIVERGENCE"
>
> > [Q5] Did nnBD-LSIF face any of the claimed issues in the second experiment?
>
> [A5]
>
> No, nnBD-LSIF did not encounter the issue in the experiment.
> Since nnBD-LSIF, like our proposed method, utilizes an unbiased gradient, the issue discussed in Section 3.3, titled "BIASED GRADIENT PROBLEM", is not expected to occur. This was confirmed by the experiment presented in Section 6.2, titled "EXPERIMENTS ON THE IMPROVEMENT OF OPTIMIZATION EFFICIENCY BY REMOVING GRADIENT BIAS"

---

> ### Author Response · Authors · 2024-12-02
>
> > Theory:
>
> > Biased gradients
> >> [Q6] * I did not immediately see why the gradient for KL is biased, can you show this formally? Biased gradients are only claimed for KL, what about other f-divergences?
>
> [A6]
>
> To address your question, we demonstrate why the gradient for the KL divergence loss is biased by showing that the expectation of the gradient of the KL divergence loss is not equal to the gradient of the expected loss.
>
> Let $\phi\_{\theta} =  |x - \theta| $, where $x \in [0,1]$ and $\theta \in (0,1)$.
> The loss function is defined as: $\mathcal{L}\_{KL}(\phi\_{\theta}) = - \hat{E}_Q\big[\log \phi\_{\theta}\big] + \hat{E}_P\big[\phi\_{\theta}\big] - 1$, and the gradient is expressed as $\nabla\_{\theta} \mathcal{L}\_{KL}(\phi\_{\theta}) = -\hat{E}_Q\big[ \nabla\_{\theta}(\log \phi\_{\theta})\big] + \hat{E}_P\big[\nabla\_{\theta}(\phi\_{\theta})\big]$.
>
> The expectation of the loss is:
>
> $$
>  E\big[ \mathcal{L}\_{KL}(\phi\_{\theta})\big] = - E_Q\big[\log \phi\_{\theta}\big] + E_P\big[\phi\_{\theta}\big] - 1,
> $$
>
> and the expectation of the gradient is:
>
> $$
>  E\big[ \nabla\_{\theta} \mathcal{L}\_{KL}(\phi\_{\theta})\big] = - E_Q\big[ \nabla\_{\theta}(\log \phi\_{\theta})\big] + E_P\big[\nabla\_{\theta}(\phi\_{\theta})\big].
> $$
>
> Now, we compute
> $$
> \frac{\partial}{\partial \theta } E\big[\log \phi\_{\theta}(x)\big] = \frac{\partial}{\partial \theta } \int\_{0}^{1} \log |x - \theta| \, dx = - \log(1 - \theta).
> $$
> while:
>
> $$E\bigg[\frac{\partial}{\partial \theta } \log  \phi\_{\theta}(x)   \bigg] =  \int\_{0}^{\theta} \frac{1}{\theta-x} dx   + \int\_{\theta}^{1} \frac{1}{x-\theta} dx = \infty.
> $$
>
> Thus, $\nabla\_{\theta} E_Q\big[\log \phi\_{\theta}\big] \neq E_Q\big[\nabla\_{\theta}(\log \phi\_{\theta})\big]$.
>
> we observe that:
>
> $$
> \nabla\_{\theta}  E\big[ \mathcal{L}\_{KL}(\phi\_{\theta})\big] \neq E\big[ \nabla\_{\theta} \mathcal{L}\_{KL}(\phi\_{\theta})\big].
> $$
>
>
> The phenomenon of biased gradients is not exclusive to the KL divergence. For instance, it is expected to occur in the case of the $\alpha$-divergence loss when $\alpha < 1/2$, the squared Hellinger divergence, and the Jensen-Shannon divergence (used in vanilla GANs).
>
> ---
>
> > [Q7] * Correct me if I'm wrong, but most kernel-based methods for DRE don't suffer from biased gradients.
>
> [A7]
>
> When the parameters of kernel-based methods are optimized using minibatch optimization,
> the issue of biased gradients can arise. For example, optimization using the KL divergence loss often results in biased gradients. However, the optimization of least-squares importance fitting (LSIF) may avoid introducing such bias [1].
>
>
>
> ---
>
> > Vanishing gradients
>
> >> [Q8] * I do not really understand (3): Isn't (i) already implying vanishing gradients?
>
> [A8]
>
> We appreciate the reviewer's insightful comment and apologize for the insufficient explanation in the original manuscript. To address this, we have added further clarification in lines 167–174 of the revised manuscript.
>
> As the reviewer correctly pointed out, in Equation (3), condition (i) indeed corresponds to the vanishing gradient of the loss function.
> However, (i) alone does not guarantee the persistence of the vanishing gradient.
> Condition (ii), therefore, plays a crucial role in ensuring the persistence of (i).
> To illustrate this, let us consider the case of the KL-divergence loss.
>
> The expected gradients of the loss function are given by:
> $$
> E\big[\nabla\_{\theta} \mathcal{L}\_{KL}(\phi\_{\theta})\big] = - E_Q\big[\nabla\_{\theta} \phi\_{\theta} / \phi\_{\theta}\big] + E_P\big[\nabla\_{\theta} \phi\_{\theta}\big].
> $$
> From this expression, condition (ii) (i.e., $\nabla\_{\theta} \phi\_{\theta} = \vec{0}$) cannot generally be derived from the requirement that $E\big[\nabla\_{\theta} \mathcal{L}\_{KL}(\phi\_{\theta})\big] = \vec{0}$. Thus, condition (i) does not necessarily imply condition (ii).
>
> Additionally, since condition (ii) signifies that no updates to the model parameters occur, the model predictions unchanged under (ii). Thus, (i) can remain under (ii).
>
> Therefore, when both (i) and (ii) are satisfied, the vanishing gradient of the loss function persists.

---

> ### Author Response · Authors · 2024-12-03
>
> > [Q9] * For me it is not clear how the limits in Table 2 are computed, a more formal derivation is needed.
>
> [A9]
>
> Thank you for pointing out the need for a clearer explanation regarding the derivation of the limits in Table 2. Below, we provide a formal derivation using the example of the Pearson $\chi^2$ divergence:
>
> The loss function is expressed as
>
> $$
> \nabla\_{\theta} L_f(\phi\_{\theta}) = -2 \cdot \hat{E}_Q[\nabla\_{\theta} \phi\_{\theta}] + 2 \cdot \hat{E}_P[\nabla\_{\theta} \phi\_{\theta} \cdot \phi\_{\theta}].
> $$
>
> Considering the case where $\phi\_{\theta} \to 0$, we analyze the limit of the gradient:
>
> $$
> \lim\_{\phi\_{\theta} \to 0} \nabla\_{\theta} L\_f(\phi\_{\theta}) = \lim\_{\phi\_{\theta} \to 0} \bigg\\{ -2 \cdot \hat{E}\_Q\big[\nabla\_{\theta} \phi\_{\theta}\big] + 2 \cdot \hat{E}\_P\big[\nabla\_{\theta} \phi\_{\theta} \cdot \phi\_{\theta}\big] \bigg\\}.
> $$
>
> This can be rewritten as
>
> $$
> \lim\_{\phi\_{\theta} \to 0} \nabla\_{\theta} L_f(\phi\_{\theta}) = -2 \cdot \hat{E}_Q\big[\nabla\_{\theta} \phi\_{\theta}\big] + 2 \cdot \lim\_{\phi\_{\theta} \to 0} \hat{E}_P\big[\nabla\_{\theta} \phi\_{\theta} \cdot \phi\_{\theta}\big].
> $$
>
> Considering the expectation, $E[\cdot] = E_P[E_Q[\cdot]]$, we obtain
>
> $$
> E\left[\lim\_{\phi\_{\theta} \to 0} \nabla\_{\theta} L_f(\phi\_{\theta})\right] = -2 \cdot E_Q[\nabla\_{\theta} \phi\_{\theta}] + 2 \cdot E\left[\lim\_{\phi\_{\theta} \to 0} \hat{E}_P[\nabla\_{\theta} \phi\_{\theta} \cdot \phi\_{\theta}]\right].
> $$
>
> Since $\nabla\_{\theta} \phi\_{\theta} \cdot \phi\_{\theta} \to 0$ as $\phi\_{\theta} \to 0$, the term $\hat{E}_P[\nabla\_{\theta} \phi\_{\theta} \cdot \phi\_{\theta}]$ vanishes: $\hat{E}_P[\nabla\_{\theta} \phi\_{\theta} \cdot \phi\_{\theta}] = 0$
>
> Thus, we have
>
> $$
> E\left[\lim\_{\phi\_{\theta} \to 0} \hat{E}_P[\nabla\_{\theta} \phi\_{\theta} \cdot \phi\_{\theta}]\right] = 0.
> $$
>
> From the above, it follows that
>
> $$
> \lim\_{\phi\_{\theta} \to 0} E[\nabla\_{\theta} L_f(\phi\_{\theta})] = -2 \cdot E_Q[\nabla\_{\theta} \phi\_{\theta}].
> $$
>
> Therefore, the results of the Pearson $\chi^2$ divergence corresponds to the second row of Table 2.
>
>
> We hope this clarifies the formal derivation. Please feel free to reach out if further explanation is needed.
>
> ---
>
> > [Q10] * Why should (3) happen during training?
>
> [A10]
>
> Equation (3) provides a sufficient condition for the gradient to remain vanished. While other conditions might also lead to this outcome, this particular condition, as explained in [A8], can be interpreted as a general condition for ensuring the persistence of the vanishing gradient phenomenon.
>
> ---
>
> >  [Q11] Why do very small/large density ratios imply (3), ...?
>
> [A11]
>
> The cases of very small or very large density ratios represent common conditions under which situation (3) can arise across $f$-divergence loss functions. While other conditions may also lead to situation (3), we have not identified any that consistently produce situation (3) for all $f$-divergence loss functions.
>
> ---
>
> > [Q12] in particular how does this determine the gradient in parameter space?
>
> [A12]
>
> The cases of extremely small or large density ratios do not directly determine the values of parameter gradients. Instead, they provide conditions under which the model gradients are constrained to remain near zero.
>
> In other words, even when the density ratios are very small or very large, it is possible for the model gradient to be nonzero. However, once the model gradient becomes zero, this state (i.e., the vanishing of the gradient) is expected to persist.
>
> ---
>
> > [Q13] * Please explain the correctness of the two equivalences in line 347 in more detail.
>
> [A13]
>
> We apologize for the lack of detail in our previous explanation. Below, we provide a more detailed discussion regarding $ \lim\_{k \to \infty} E_Q[e^{T_k}] \to 0 \iff \lim\_{k \to \infty} E_P[e^{T_k}] \to 0 $.
>
> Let $ A = \\{\mathbf{x} \in \Omega : \lim\_{k \to \infty} e^{T_k} = 0 \\} $.
>
> Here, $ k $ represents the number of training steps.
>
> When $ \lim\_{k \to \infty} E_Q[e^{T_k}] \to 0 $, it follows that $ \lim\_{k \to \infty} e^{T_k} = 0 $ almost everywhere (a.e.) under $ Q $. Hence, $ Q(A) = 0 $. Since $ p(\mathbf{x}) > 0 \iff q(\mathbf{x}) > 0 $, we have $ P(A) = 0 $. Therefore, $ \lim\_{k \to \infty} e^{T_k} = 0 $ a.e. under $ P $. Consequently, the following holds:
>
> $$
> \lim\_{k \to \infty} E_P[e^{T_k}] = \lim\_{k \to \infty} E_P[|e^{T_k}|] = E_P[\lim\_{k \to \infty} |e^{T_k}|] = E_P[\lim\_{k \to \infty} e^{T_k}] = 0.
> $$

---

> ### Author Response · Authors · 2024-12-03
>
> > * [Q14] Please show the computation of the limits in Table 4 in more detail.
>
> [A14]
>
> These results can be derived similarly to those in **[A10]**. Below, we detail the case of $\alpha < 0$, where $E_P[e^{T\_{\theta}}] \to \infty$ in $\nabla\_{\theta} \mathcal{L}\_{\alpha\text{-Div}}(T\_{\theta})$.
>
> Recall Equation (13):
>
> $$
> \nabla\_{\theta} \mathcal{L}\_{\alpha\text{-Div}}(T\_{\theta}) = \hat{E}_Q[\nabla\_{\theta} T\_{\theta} \cdot e^{\alpha \cdot T\_{\theta}}] - \hat{E}_P[\nabla\_{\theta} T\_{\theta} \cdot e^{(\alpha - 1) \cdot T\_{\theta}}].
> $$
>
> Considering the limit as $e^{T\_{\theta}} \to \infty$, we have:
>
> $$
> \lim\_{e^{T\_{\theta}} \to \infty} \nabla\_{\theta} \mathcal{L}\_{\alpha\text{-Div}}(T\_{\theta}) = \lim\_{e^{T\_{\theta}} \to \infty} \hat{E}_Q[\nabla\_{\theta} T\_{\theta} \cdot e^{\alpha \cdot T\_{\theta}}] - \lim\_{e^{T\_{\theta}} \to \infty} \hat{E}_P[\nabla\_{\theta} T\_{\theta} \cdot e^{(\alpha - 1) \cdot T\_{\theta}}].
> $$
>
> This can be rewritten as:
>
> $$
> \hat{E}_Q\left[\lim\_{e^{T\_{\theta}} \to \infty} \nabla\_{\theta} T\_{\theta} \cdot e^{\alpha \cdot T\_{\theta}}\right] - \hat{E}_P\left[\lim\_{e^{T\_{\theta}} \to \infty} \nabla\_{\theta} T\_{\theta} \cdot e^{(\alpha - 1) \cdot T\_{\theta}}\right]. \tag{B3}
> $$
>
> Since $\alpha < 0$ and $\alpha - 1 < 0$, we observe that:
>
> $$
> \lim\_{e^{T\_{\theta}} \to \infty} e^{\alpha \cdot T\_{\theta}} = 0 \quad \text{and} \quad \lim\_{e^{T\_{\theta}} \to \infty} e^{(\alpha - 1) \cdot T\_{\theta}} = 0.
> $$
>
> Substituting these results into Equation (B3), we get:
>
> $$
> \hat{E}_Q\left[\lim\_{e^{T\_{\theta}} \to \infty} \nabla\_{\theta} T\_{\theta} \cdot e^{\alpha \cdot T\_{\theta}}\right] - \hat{E}_P\left[\lim\_{e^{T\_{\theta}} \to \infty} \nabla\_{\theta} T\_{\theta} \cdot e^{(\alpha - 1) \cdot T\_{\theta}}\right] = 0. \tag{B4}
> $$
>
> Thus:
>
> $$
> \lim\_{e^{T\_{\theta}} \to \infty} \nabla\_{\theta} \mathcal{L}\_{\alpha\text{-Div}}(T\_{\theta}) = 0,
> $$
>
> which corresponds to the result shown in the second-to-last column of the first row in Table 4.
>
> ---
>
> > [Q15] * A bad sample rate is only shown for KL-divergence, what about the other f-divergences?
>
> [A15]
>
> Thank you for your insightful question.
>
> First, we would like to clarify the distinction between the types of convergence presented in Theorem 4.2 of this study and those discussed in [4].
>
> Theorem 4.2 in our study focuses on the convergence rate of the variance of the estimator, as well as the convergence rate of the KL-divergence estimators provided in Equation (4).
> In contrast, the sample rates for DRE described in [4] pertain to the convergence rate of the bias of the $f$-divergence estimators. Specifically, [4] provides convergence rates for the upper bounds of $BF(\beta, g(f\_{\lambda, t, z})) - BF(\beta, g(f_H))$. On the other hand, our work presents an upper bound for $\big\{BF(\beta, g(f\_{\lambda, t, z})) - BF(\beta, g(f_H))\big\\}^2$.
>
> Thus, we found it difficult to directly compare the sample rates for DRE derived in Theorem 4.2 of our study with those in [4], and we would like to address the reviewer's question regarding the sample rate for KL-divergence estimation by briefly reviewing the sample rates for the variance of the $f$-divergence estimators.
>
> - In general, the central limit theorem ensures $\sqrt{N}$-consistency. An example where explicit rates for the Hellinger divergence are derived can be found in [B1].
>
> - To the best of our knowledge, prior studies have only established a relationship between the KL-divergence of data and the sample size requirements for KL-divergence estimation.
>
>
> [B1] Birrell, J., Katsoulakis, M. A., & Pantazis, Y. (2022). Optimizing variational representations of divergences and accelerating their statistical estimation. IEEE Transactions on Information Theory, 68(7), 4553-4572.

---

> ### Author Response · Authors · 2024-12-03
>
> >  [Q16] * Describe the implications of Theorem 6.2 for the sample rate in more detail.
>
> [A16]
>
> Thank you for your insightful comment. Below, we elaborate on the implications of Theorem 6.2 for the sample rate in greater detail.
>
> According to Theorem 6.2, the product of the variance of the estimated $\alpha$-divergence and $\sqrt{N}$ is bounded by a constant that is independent of the value of the divergence when $\alpha$ is fixed.
>
> In particular, when $\alpha = \frac{1}{2}$, Equation (16) specifies the variance $\sigma\_{\alpha}^2$ as:
> $$
> \sigma\_{\alpha}^2 = -8 \cdot D(Q \| P; 1/2)^2 + 16 \cdot D(Q \| P; 1/2) + 12 \leq 20.
> $$
>
> From this result, the 95% confidence interval for the estimated $\alpha$-divergence can be calculated. Then, the width of this interval is bounded by:
> $$
> \frac{4 \cdot \sqrt{20}}{\sqrt{N}}.
> $$
> Thus, this interval is independent of the true value of the $\alpha$-divergence.
>
> In contrast, for the KL-divergence, as described in Equation (4), the width of the confidence interval depends directly on the true value of the KL-divergence. Specifically, it is lower-bounded by:
> $$
> 4 \cdot (e^{KL(Q \| P)} - 1).
> $$
>
> This distinction highlights an advantage of the $\alpha$-divergence in divergence:
> The sample requirement for maintaining a fixed confidence interval width remains constant regardless of the underlying divergence value, unlike the KL-divergence, where the sample rate may need to increase significantly with larger divergence values.

---

### Official Review · Reviewer_2nWh · 2024-11-03

**Soundness:** 1
**Presentation:** 2
**Contribution:** 2
**Rating:** 3
**Confidence:** 4

**Summary:**

This work studies the problem of density ratio estimation (DRE), i.e., estimating the ratio $p(x)/q(x)$ using neural networks, where both $p$ and $q$ are probability densities. The authors argue that the common approach of optimizing a variational formulation of the KL-divergence (whose optimum is $p/q$) has a number of difficulties, namely: overfitting, biased and vanishing gradients, and high effective sample size. The authors then consider an alternative formulation based on Amari's $\alpha$-divergences instead, and develop a parametrization which does not suffer from the same problem as the original one. The author demonstrate these properties empirically, yet they also show that in terms of regression error, there is no significant advantage of using their method instead of the KL-divergence loss.

**Strengths:**

- **S1.** The problem addressed by the authors is of great interest to a part of the ML community.
- **S2.** The authors gather a good summary of the problems that existing approaches have.
- **S3.** The authors make a great effort into explaining the details of their proposed loss, which looks like a sensible approach.
- **S4.** While I have only skimmed through them, the results in the appendix on real-world data looks good, and I am not sure why it is not in the main manuscript.

**Weaknesses:**

- **W1.** The presentation could be improved, e.g., there are some clear typos that should be corrected ("hucking" instead of "hacking" in section 6.1, "Peason" in Table 1, "simbole" in line 231).
- **W2.** Some of the "intuitive explanations" of the paper are wrong or are questionable at best. For example:
  1. The problem of train-loss hacking is attributed here to the loss not being lower-boundered, but I'd say that the problem is explained through the fact that there is a finite number of training samples. You could still lower-bound the loss, if the lower-bound is only tight at the limit, use the same argument as in section 4.2.
  2. Line 161 is wrong. Under mild conditions, the linearity of the integral allows you to swap the integral and gradient, as long as $Q$ does not depend on $\theta$, which I understand is the case.
  3. The arguments in section 4.4 sound cyclic, i.e., arguments (i) and (ii) of lines 181-184 are equivalent to the gradients vanishing.
  4. (more nitpicky, but:) The assumption in the appendix for the theorem 6.1 is local L-continuity, which is worth mentioning in the main paper.
- **W3.** I have found several errors and questionable explanations in the manuscript, which make me worry about whether the results presented in the manuscript hold after fixing them. For example:
  1. Line 269 is false under fairly general assumptions. You can swap gradients and integral as long as Q does not depend on $\theta$ (or if Q is reparametrizable).
  2. The last two columns of Table 2 are all wrong as far as I can see ($E$ should be $\hat E$ and the $Q$ in the last column should be $P$.)
  3. While not "wrong" per se, as it does not change the arguments of the paper, writing a difference between vector infinities in Table 4 is rather concerning.
  4. I quickly skimmed through the proofs, and I do not understand Eq. 57 at all. Are the authors writing a gradient as the limit of a function as $\psi$ approaches $\theta$? Where is this definition coming from?
- **W4.** The baseline methods to compare their approach with change across experiments, which I cannot fully undestand. Why not using nnBD-LSIF for the experimerints in section 7.3?
- **W5.** This is more of a personal taste, but I really dislike some of the wording chosen in this manuscript, e.g., using "naive" and "merely" as freely as in lines 48-49. It downplays the contribution of previous works in an unfair manner.

**Questions:**

See above.

---

> ### Author Response · Authors · 2024-12-03
>
> Due to personal circumstances, we required some time to prepare this response, and we sincerely apologize for the delay in addressing your inquiry.
>
> We fully understand that this delay may leave limited time for you to review our response. Nonetheless, as a token of our gratitude for the time and effort you have invested in reviewing our work, we have provided detailed answers to the questions you raised.
>
>
> ---
>
> > [Q1] W1. The presentation could be improved, e.g., there are some clear typos that should be corrected ("hucking" instead of "hacking" in section 6.1, "Peason" in Table 1, "simbole" in line 231).
>
> [A1]
>
> We sincerely apologize for the numerous typographical errors. We have made corrections in the revised version.
>
> ---
>
> > W2. Some of the "intuitive explanations" of the paper are wrong or are questionable at best.
>
> >> [Q2-1] 1. For example: The problem of train-loss hacking is attributed here to the loss not being lower-boundered, but I'd say that the problem is explained through the fact that there is a finite number of training samples. You could still lower-bound the loss, if the lower-bound is only tight at the limit, use the same argument as in section 4.2.
>
> [A2-1]
>
> We apologize for the lack of clarity and insufficient explanation in the initial submission. In the revised version, we have added more detailed discussions on gradient vanishing (Section 3.3) and unbiased gradients (Section 3.4) to address these concerns comprehensively.
>
> While we appreciate the reviewer's insightful suggestion, we respectfully differ in our interpretation of the root cause of the train hacking problem associated with unbounded loss functions. Specifically, we do not view it as fundamentally an issue of overfitting due to a finite number of training samples.
>
> As the reviewer correctly noted, training neural networks on datasets from two distinct domains can indeed lead to overfitting. However, based on the results of the numerical experiments presented in Section 6.1, the train hacking problem appears to be a distinct phenomenon that arises exclusively with lower-unbounded loss functions. Furthermore, this issue manifests earlier and more acutely than general overfitting typically does.
>
> We therefore propose that the train hacking problem is best understood as a specific form of overfitting, where the training process disproportionately accelerates in the unbounded term of the $f$-divergence loss function compared to other terms.
>
> ---
>
> >>  [Q2-2] Line 161 is wrong. Under mild conditions, the linearity of the integral allows you to swap the integral and gradient, as long as  does not depend on , which I understand is the case.
>
> [A2-2]
> We believe that it is challenging to assume mild and natural conditions for the interchangeability of the expectation operator $E$ and the (partial or total) derivative of the KL-divergence loss for the following reasons:
>
> As addressed in our response to Reviewer Htnn's question [Q6], the interchangeability of the integral and gradient for the KL-divergence fails in standard estimation problems, such as the estimation of a shift parameter.
>
> From a theoretical perspective, the interchangeability of $E$ and the differential operator $\nabla$ generally requires the condition of the uniform integrability. It is important to note that the uniform integrability of the KL-divergence loss, particularly terms expressed as "$\log |\cdot|$", does not hold when the integration domain includes intervals around zero.

---

> ### Author Response · Authors · 2024-12-03
>
> >> [Q2-3] 3. The arguments in section 4.4 sound cyclic, i.e., arguments (i) and (ii) of lines 181-184 are equivalent to the gradients vanishing.
>
> [A2-3]
>
> We sincerely thank the reviewer for their insightful comment and apologize for the insufficient explanation provided in the original manuscript. In response, we have added further clarification in lines 167–174 of the revised manuscript.
>
> To directly address the reviewer's question, below we reiterate the explanation provided in response to review Htnn's question [Q8], as it is relevant to this matter.
>
> As the reviewer correctly pointed out, in Equation (3), condition (i) indeed corresponds to the vanishing gradient of the loss function.
> However, (i) alone does not guarantee the persistence of the vanishing gradient.
> Condition (ii), therefore, plays a crucial role in ensuring the persistence of (i).
> To illustrate this, let us consider the case of the KL-divergence loss.
>
> The expected gradients of the loss function are given by:
> $$
> E\big[\nabla\_{\theta} \mathcal{L}\_{KL}(\phi\_{\theta})\big] = - E_Q\big[\nabla\_{\theta} \phi\_{\theta} / \phi\_{\theta}\big] + E_P\big[\nabla\_{\theta} \phi\_{\theta}\big].
> $$
> From this expression, condition (ii) (i.e., $\nabla\_{\theta} \phi\_{\theta} = \vec{0}$) cannot generally be derived from the requirement that $E\big[\nabla\_{\theta} \mathcal{L}\_{KL}(\phi\_{\theta})\big] = \vec{0}$. Thus, condition (i) does not necessarily imply condition (ii).
>
> Additionally, since condition (ii) signifies that no updates to the model parameters occur, the model predictions unchanged under (ii). Thus, (i) can remain under (ii).
>
> Therefore, when both (i) and (ii) are satisfied, the vanishing gradient of the loss function persists.
>
> ---
>
> > [Q2-4] (more nitpicky, but:) The assumption in the appendix for the theorem 6.1 is local L-continuity, which is worth mentioning in the main paper.
>
> [A2-4]
>
> Thank you for pointing this out and for your constructive feedback.
> We have added this description to line 311 of the revised manuscript to address your comment.
>
> ---
>
> > W3. I have found several errors and questionable explanations in the manuscript, which make me worry about whether the results presented in the manuscript hold after fixing them. For example:
>
> >> [Q3-1] 1. Line 269 is false under fairly general assumptions. You can swap gradients and integral as long as Q does not depend on  (or if Q is reparametrizable).
>
> [A3-1]
>
> As addressed in our response to [A2-2], the exchangeability of the expectation operator $ E $ and the (partial or total) differential operator $ \nabla $ generally requires the uniform integrability of the loss function.
>
> In the case of the terms in the $\alpha$-divergence loss, $ \phi_{\theta}^{\alpha} $, the condition of uniform integrability is not satisfied when the integration domain includes intervals around zero and $\alpha < \frac{1}{2}$.
>
> To illustrate this, consider the following example:
> Let $ f_{\theta}(x) = |x - \theta|^{-\frac{1}{3}} $, where $ x \in [0, 1] $ and $ \theta \in (0, 1) $.
>
> We compute the expectation:
>
> $$
> E[f_{\theta}(x)] = \int_0^1 f_{\theta}(x) \, dx = \frac{1}{2} \cdot (1 - \theta)^{-\frac{1}{3}} + \theta^{-\frac{1}{3}},
> $$
>
> and the expectation of the derivative:
>
> $$
> E\left[\frac{\partial f_{\theta}(x)}{\partial \theta}\right] = \int_0^1 -\frac{1}{3} |x - \theta|^{-\frac{4}{3}} \, dx = \infty.
> $$
>
> Thus, we observe the following inequality:
>
> $$
> \frac{\partial}{\partial \theta} E[f_{\theta}(x)] \neq E\left[\frac{\partial f_{\theta}(x)}{\partial \theta}\right].
> $$
>
> This example demonstrates that under the given conditions, the exchangeability of the expectation and the differential operators fails to hold.

---

> ### Author Response · Authors · 2024-12-03
>
> > [Q3-2] The last two columns of Table 2 are all wrong as far as I can see ( should be  and the $Q$ in the last column should be $P$.)
>
> [A3-2]
> We confirm that the $E$ values in Table 2 are correct, as they are consistent with Equation (3) and align with the notation used in the discussion under "5.3 ADDRESSING GRADIENT VANISHING PROBLEM."
>
> While it is true that $\hat{E}$ could also be valid, we believe that $E$ is more appropriate in this context because it represents a statistical property.
>
> However, we acknowledge that the $Q$ in the last column should indeed be $P$. We sincerely thank the reviewer for bringing this oversight to our attention. We apologize for not addressing this issue during the revision process, as we did not have sufficient time for a comprehensive verification.
>
>
> > [Q3-3] While not "wrong" per se, as it does not change the arguments of the paper, writing a difference between vector infinities in Table 4 is rather concerning.
>
> [A3-3]
>
> Thank you for your constructive feedback regarding the notation in Table 4.
>
> In the revised manuscript, we have updated the description to adopt a more appropriate and clear style. Specifically, we now use $\infty$ instead of $\overset{\rightarrow}{\infty}$, aligning with the use of the max norm $\\|\cdot\\|_{\infty}$.
>
> We hope this adjustment addresses your concern effectively.
>
> ---
>
> > [Q3-4] I quickly skimmed through the proofs, and I do not understand Eq. 57 at all. Are the authors writing a gradient as the limit of a function as approaches? Where is this definition coming from?
>
> [A3-4]
>
> Thank you for pointing this out.
> As the reviewer correctly noted, $T$ in Equation (57) should be $T_{\theta}$, as the subscript $\theta$ was missing.
>
> We sincerely apologize for the confusion caused by this oversight. This issue has been corrected in the revised version of the manuscript.
>
> ---
>
> > [Q4] W4. The baseline methods to compare their approach with change across experiments, which I cannot fully undestand. Why not using nnBD-LSIF for the experimerints in section 7.3?
>
> [A4]
>
> We opted to use the $f$-divergence loss instead of nnBD-LSIF for the experiments in Section 7.3 because nnBD-LSIF is not equivalent to the variational expression of the $f$-divergence.
>
> Specifically, nnBD-LSIF includes an additional term that is not derived from the variational expression of divergence. This term is introduced to prevent the negative divergence of its lower-unbounded component. However, the influence of this additional term on the optimization process remains unclear, making it challenging to fully understand its impact on divergence minimization.
>
> To ensure consistency and clarity, we employed an $f$-divergence loss directly derived from its variational expression. This approach guarantees that the optimization process aligns precisely with the optimization of the $f$-divergence itself.
>
> ---
>
> > [Q5] This is more of a personal taste, but I really dislike some of the wording chosen in this manuscript, e.g., using "naive" and "merely" as freely as in lines 48-49. It downplays the contribution of previous works in an unfair manner.
>
> [A5]
>
> We sincerely apologize for any unintended impression of downplaying prior work caused by the wording in our manuscript. We appreciate your feedback and have revised the expressions to better reflect our respect for previous research. Specifically, we have made the following changes:
>
> - **"naive"** → **"standard"**
> - **"merely"** → **"directly"**
>
> These adjustments ensure that our manuscript conveys its intended meaning without diminishing the contributions of earlier studies. Thank you for bringing this to our attention.

---

### Author Response · Authors · 2024-11-28

Thank you very much for taking the time to review our submission and for providing detailed and constructive feedback.

We sincerely apologize for the significant delay in providing our response regarding this matter.

Based on the feedback we received, we have revised the manuscript and submitted the updated version.

These changes were aimed at improving the clarity of the discussion while keeping the overall content of the study unchanged.

We have revised the manuscript in the following points:
1. We have added an explanation regarding the replacement of integrals with derivatives (Section 3.3 of the revised manuscript, BIASED GRADIENT PROBLEM, lines 147–149).
2. We have added a detailed discussion regarding conditions for the vanishing of the gradient (Section 3.4 of the revised manuscript, VANISHING GRADIENTS PROBLEM, lines 167–173).
3. We have added the Related Work description in Section B in the appendix (due to the limitations of space).
4. We have revised impolite expressions (e.g., "naive" → "standard," "merely" → "directly").
5. We have corrected typographical errors.


I also plan to address the questions raised by each reviewer. However, given the limited time available, I may ask the reviewers to wait until I have provided comprehensive responses to their questions.

Considering the significant delay in my replies, please feel free to disregard my answers.
However, we would deeply appreciate it if the reviewers could kindly take a look at the updated discussion.

Thank you for your understanding, and I apologize for the delay in my response to your questions.

Best regards,

Authors of Submission 1919

---

### Meta-Review · Area_Chair_QtRB · 2024-12-19

**Metareview:**

This paper addresses the problem of density ratio estimation (DRE). The authors propose to use \alpha-divergence loss so that several problems (i.e., bias, overfitting, vanishing gradient) in standard methods can be resolved.

The problem itself is important. This paper addresses an interesting problem. However, the writing can be much improved. There are typos and unclear logics. Comparison with existing work is not thoroughly given. These issues could be addressed but a substantial revision will be required.

For these reasons, this paper cannot be accepted.

**Additional Comments On Reviewer Discussion:**

The authors responded carefully to the reviewers' concerns. However, they require substantial change of the paper. Therefore, the evaluation was not changed.

---

### Decision · Program_Chairs · 2025-01-22

Reject